# The translocation of a chloride channel from the Golgi to the plasma membrane helps plants adapt to salt stress

Sivamathini Rajappa [1,9], Pannaga Krishnamurthy [1,2,9], Hua Huang[3,4,5,6], Dejie Yu[3,4,5,6], Jiří Friml [7], Jian Xu[8] & Prakash P. Kumar [1,2] ✉

A key mechanism employed by plants to adapt to salinity stress involves maintaining ion homeostasis via the actions of ion transporters. While the function of cation transporters in maintaining ion homeostasis in plants has been extensively studied, little is known about the roles of their anion counterparts in this process. Here, we describe a mechanism of salt adaptation in plants. We characterized the chloride channel (*CLC*) gene *AtCLCf*, whose expression is regulated by WRKY transcription factor under salt stress in *Arabidopsis thaliana*. Loss-of-function *atclcf* seedlings show increased sensitivity to salt, whereas *AtCLCf* overexpression confers enhanced resistance to salt stress. Salt stress induces the translocation of GFP-AtCLCf fusion protein to the plasma membrane (PM). Blocking AtCLCf translocation using the exocytosis inhibitor brefeldin-A or mutating the small GTPase gene *AtRABA1b/BEX5* (*RAS GENES FROM RAT BRAINA1b* homolog) increases salt sensitivity in plants. Electrophysiology and liposome-based assays confirm the Cl⁻/H⁺ antiport function of AtCLCf. Therefore, we have uncovered a mechanism of plant adaptation to salt stress involving the NaCl-induced translocation of AtCLCf to the PM, thus facilitating Cl⁻ removal at the roots, and increasing the plant's salinity tolerance.

Increased soil salinity is a growing challenge for crop production, which adversely affects crop yields in ~20% of cultivable land worldwide. This problem is predicted to affect >40% of agricultural lands in the coming decades[1]. The excess accumulation of soluble salts, especially sodium chloride (NaCl), in the root zone severely impedes plant growth, reducing crop productivity. Previous efforts to identify solutions for mitigating salinity stress have focused primarily on cations such as sodium (Na⁺)[2]. It is important to explore the effects of counter-anions such as chloride (Cl⁻) on root growth to help plants cope with increased soil salinity. Although Cl⁻ is an essential nutrient for crop plants and is used as osmoticum in halophytes[3,4], excess Cl⁻ inhibits plant growth and yield[5]. Cl⁻ efflux from roots is positively correlated with salt tolerance[6], suggesting that it might be the key mechanism preventing Cl⁻ toxicity in plants.

[1]Department of Biological Sciences and Research Centre on Sustainable Urban Farming, National University of Singapore, 14 Science Drive 4, Singapore 117543, Singapore. [2]NUS Environmental Research Institute, National University of Singapore, #02-01, T-Lab Building, 5A Engineering Drive 1, Singapore 117411, Singapore. [3]Department of Physiology, Yong Loo Lin School of Medicine, National University of Singapore, Singapore 117597, Singapore. [4]Electrophysiology Core Facility, Yong Loo Lin School of Medicine, National University of Singapore, Singapore 117456, Singapore. [5]Healthy Longevity Translational Research Program, Yong Loo Lin School of Medicine, National University of Singapore: Level 5, Centre for Life Sciences, 28 Medical Drive, Singapore 117456, Singapore. [6]Cardiovascular Diseases Program, National University of Singapore, 14 Medical Drive, MD6, #08-01, Singapore 117599, Singapore. [7]Institute of Science and Technology Austria (IST Austria) Am Campus 1, 3400 Klosterneuburg, Austria. [8]Department of Plant Systems Physiology, Radboud Institute for Biological and Environmental Sciences, Radboud University, Huygens Building, Heyendaalseweg 135, 6500 AJ Nijmegen, The Netherlands. [9]These authors contributed equally: Sivamathini Rajappa, Pannaga Krishnamurthy. ✉e-mail: prakash.kumar@nus.edu.sg

However, little is known about how roots carry out Cl⁻ efflux to overcome salinity stress.

Recent studies have implicated plant chloride channels (CLCs) as involved in regulating Cl⁻ homeostasis under salinity stress[7,8]. CLCs play vital roles in stabilizing membrane potential and turgor pressure. These integral membrane proteins contain 10–12 transmembrane domains and are present in all kingdoms of life[9,10]. Although the functions of plant CLCs are not well-studied, extensive information exists for their animal counterparts, revealing important effects in both normal development and disease states. For example, mutations in mammalian *CLC* genes cause genetic diseases such as Bartter syndrome, idiopathic epilepsy, osteopetrosis, Dent's disease, and various types of myopathies[11,12]. However, the effects of *CLC* mutations in plants are unclear.

Despite their nomenclature, CLC family members are not exclusively Cl⁻ transporters. Although some CLCs function as Cl⁻ channels or chloride-proton antiporters ($Cl^-/H^+$), others act as $NO_3^-$ transporters[10,13]. The transmembrane regions of $Cl^-/H^+$ antiporter CLCs are characterized by two key conserved 'gating' and 'proton' glutamate residues, whereas the Cl⁻ channels contain only the gating glutamate in the selectivity filter domain[9,14]. Among the seven CLC genes in *Arabidopsis thaliana*, we determined that *AtCLCf* is induced in roots in response to salt treatment. AtCLCf was predicted to be a Cl⁻ channel localized to the Golgi apparatus and the *trans*-Golgi network/early endosome (TGN/EE)[15,16], but how it helps reduce ion toxicity in the roots of plants under salt stress remains unknown. In addition to their function, molecular regulatory mechanism behind the expression of *CLCs* and the study on associated TFs are limited.

Here, we describe the mode of action of AtCLCf in the remediation of Cl⁻ toxicity under salt stress. Our findings show that AtCLCf expression is directly regulated by WRKY9 transcription factor and the increased intracellular NaCl levels induce the translocation of AtCLCf from the Golgi apparatus to the plasma membrane (PM) via TGN using an AtRABA1b (BEX5)-mediated pathway. The electrophysiological studies (patch clamp) with Human Embryonic Kidney 293 (HEK293FT) cells show that AtCLCf functions as a $Cl^-/H^+$ antiporter involved in efflux of Cl⁻ from the cells. The subcellular translocation of this CLC in the root cortex tissue and epidermis represents an essential salt tolerance mechanism in Arabidopsis.

## Results

### AtCLCf is essential for salt tolerance

Because roots are the primary plant organs that perceive salt stress, we examined the role of AtCLCf in root elongation using one-week-old Arabidopsis seedlings of different genotypes. The roots of *atclcf* seedlings were sensitive to 6 h of 50 mM NaCl treatment, showing significant reductions in the lengths of the meristematic zone (MZ, 20%) and elongation zone (EZ, 44%) (Fig. 1a–f) that led to shorter roots (Fig. 1g). By contrast, complemented *atclcf* (*pAtCLCf::AtCLCf;atclcf*) plants exhibited phenotypic rescue, as they grew in a similar manner to the wild type (WT) under salt stress (Fig. 1a–g), with significantly lower NaCl-induced reductions in root growth occurring in both *pAtCLCf::AtCLCf;atclcf* and *35S::AtCLCf;atclcf* plants compared to *atclcf*. Additionally, *atclcf* seedling root elongation was inhibited by KCl treatment, but not by $NaNO_3$ or the osmotic stress from mannitol treatment (Supplementary Fig. 1), suggesting that the effects seen are specific to Cl⁻ stress.

We analyzed the salt responsiveness of *AtCLCf* transcripts by performing qRT-PCR and GUS reporter assays. In the absence of NaCl treatment, roots and siliques exhibited high *AtCLCf* transcript levels (Fig. 2a–f). After 6 h of NaCl treatment, *AtCLCf* transcript levels in the roots increased ~3-fold (Fig. 2g–i and Supplementary Fig. 2a), but those in the shoots did not change significantly. These results suggest that *AtCLCf* plays an important role in maintaining Cl⁻ homeostasis in the roots. Older Arabidopsis plants showed a similar response to salt

(Fig. 2j, k): when treated with NaCl for one week, *35S::AtCLCf;atclcf* plants showed significantly higher chlorophyll content (~6 times), fresh weight (Supplementary Table 1 ~1.5 times), and survival rate (~2 times) than *atclcf* plants (Supplementary Fig. 2b, c). Severe chlorosis and stunted growth were observed in *atclcf* and WT plants in response to NaCl treatment, and these plants could not recover to the same extent as *35S::AtCLCf;atclcf* plants one week after the salt stress was withdrawn (Supplementary Fig. 3a–c). Although salt treatment increased the Cl⁻ contents in the shoots and roots of plants of all genotypes, the effect was greatest in *atclcf* shoots (Fig. 2l). These results suggest that AtCLCf functions to minimize the amount of Cl⁻ reaching the shoots, likely by removing the excess Cl⁻ at the roots, thereby protecting the photosynthetically active tissues of Arabidopsis from salinity stress.

### Perception of salinity signal by AtWRKY9 induces *AtCLCf* expression

We sought to examine how the salinity signal is transmitted to *AtCLCf*. Accordingly, we analyzed the 5'-upstream region of *AtCLCf* and detected abiotic stress-related *cis*-elements, including WRKY binding domains (Supplementary Fig. 4a). The role of WRKY TFs in perceiving salt is well-established in plants. While ectopic expression of wheat WRKYs increased salt tolerance in Arabidopsis[17], expression of several WRKYs such as WRKY9, 25, 33 was positively correlated with improved salt tolerance in other species[18,19]. Also, *AoWRKY9* was induced upon salt stress in our earlier study[20]. Interestingly, *AtCLCf* expression was significantly suppressed (~10-fold) in *atwrky9*, but not in *atwrky6* and *atwrky33* mutants (Supplementary Fig. 4b). Furthermore, ChIP-qPCR showed ~4-fold enrichment of *AtCLCf* promoter fragment by AtWRKY9-HA, while there was no significant enrichment by the other two WRKYs studied (Fig. 2m). This was supported by independent luciferase assays where ~4-fold higher luminescence was observed in *atwrky9* protoplasts transfected with *pAtCLCf::LUC* along with *35S::AtWRKY9*, compared to the control. There was no enhancement in luminescence when the WRKY binding domains in *AtCLCf* promoter fragment were mutated (Fig. 2n). The interaction between AtWRKY9 and *AtCLCf* promoter was also confirmed through Y1H assay (Supplementary Fig. 4c). Collectively, these results show that under salt stress conditions AtWRKY9 binds to *AtCLCf* promoter and enhances its expression.

### AtCLCf functions as a $Cl^-/H^+$ antiporter and is involved in Cl⁻ efflux

Some CLCs do not transport Cl⁻[15]. Hence, we tested whether AtCLCf is a functional Cl⁻ channel using liposomes incorporated with recombinant AtCLCf protein and patch clamp experiments by expressing the AtCLCf in HEK293FT cells (Fig. 3). In addition, vesicles prepared from PMs extracted from WT and *atclcf* plants treated for 6 h with NaCl were used to check the Cl⁻ transport (Supplementary Fig. 5). Channel activity was quantified as the decrease in fluorescence of a fluorophore (MQAE, loaded into the liposomes or the PM vesicles) that is quenched by Cl⁻ ions. The liposome-based fluorescence quenching assays helped to determine the transport activity of recombinant AtCLCf protein with NaCl, KCl and $CaCl_2$ as Cl⁻ donors. Data from liposomes with the incorporation of recombinant AtCLCf protein showed that the Cl⁻ transport rate was in the following order: $CaCl_2$ > NaCl > KCl, while the control liposomes without AtCLCf showed no fluorescence quenching (Fig. 3a). Increasing the concentration of NaCl in the external buffer showed a concentration dependent increase in fluorescence quenching (Fig. 3b), confirming that AtCLCf is a functional Cl⁻ channel. Similarly, the patch clamp recordings showed that, by pulsing the cells to different voltages ranging from −80 to 80 mV, AtCLCf transfected HEK293FT cells displayed significantly larger inward and outward current density as compared to vector transfected cells. (Fig. 3). The current density (pA/pF) was scored by normalizing current over cell

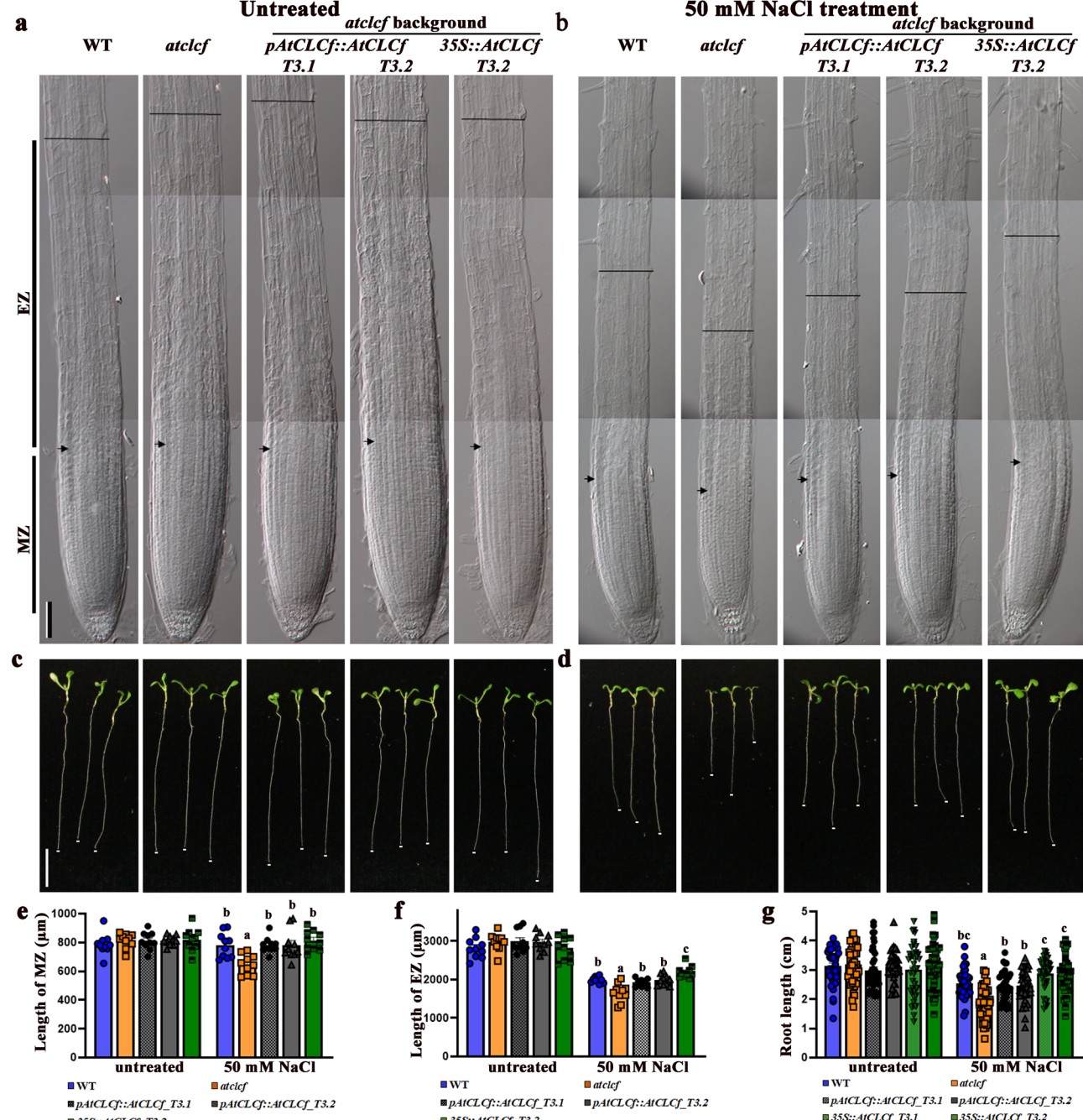

**Fig. 1 | AtCLCf is essential for salt tolerance.** DIC microscopy images of the root tips of one week-old seedlings of different genotypes grown (**a**) without and (**b**) with salt treatment (50 mM NaCl, 6 h). Arrows indicate the border between the meristematic zone (MZ) and elongation zone (EZ). Line indicates the end of the EZ near the root hair formation zone. Scale bar, 200 μm. Root lengths of seedlings (genotypes indicated at the top) grown without (**c**) and with (**d**) 50 mM NaCl treatment, at 7 days after germination; scale bar, 1 cm. Root tips are marked by white dots. Quantification of MZ (**e**) and EZ (**f**) length from **a** and **b** using ImageJ software (mean ± SE, n = 10 per genotype). **g** Root lengths of seedlings in **c** and **d**, mean ± SE, n = 40 per genotype. Means with different letters within a data set (in **e**–**g**) are significantly different, $P < 0.05$ (one-way ANOVA followed by Tukey's test). Data sets without significant differences are unlabeled. Data were obtained from three independent experiments, and representative images are shown here.

capacitance of individual cells. The data show a larger inward current ($-30.86 \pm 6.34$) than outward current ($23.45 \pm 4.54$) indicating a stronger outflux of chloride ions (Fig. 3E–G). Subsequently, to test how AtCLCf functions, we recorded the currents by changing the internal pH from 7.4 to 5.5 and found an increase in the influx of anion ($Cl^-$) suggesting an efflux of the $H^+$ ions (Supplementary Fig. 6a–c). Further, its ion selectivity was tested by checking the effect of replacing most of internal CsCl with CsF, CsBr, CsI, $CsNO_3$ or CsAcetate on the ionic current. Interestingly, there was a significant increase in $Cl^-$ efflux

among the anions tested in AtCLCf transfected cells compared to the vector (Fig. 3h–j). Concomitantly, using the pH-sensitive fluorescent dye HPTS (8-hydroxy-1,3,6-pyrenetrisulfonate) in liposome assays (Supplementary Fig. 6d, e), we observed a significant $H^+$ transport in AtCLCf-embedded liposomes when the external buffer pH was increased to 8.0 and the extravesicular halide anions were changed. Significant changes in the ion transport rate were observed, with the selectivity as: $Cl^- > F^- > I^-$. These results suggest that the $Cl^-$ transport activity of AtCLCf is coupled with $H^+$ exchange. Additionally, to

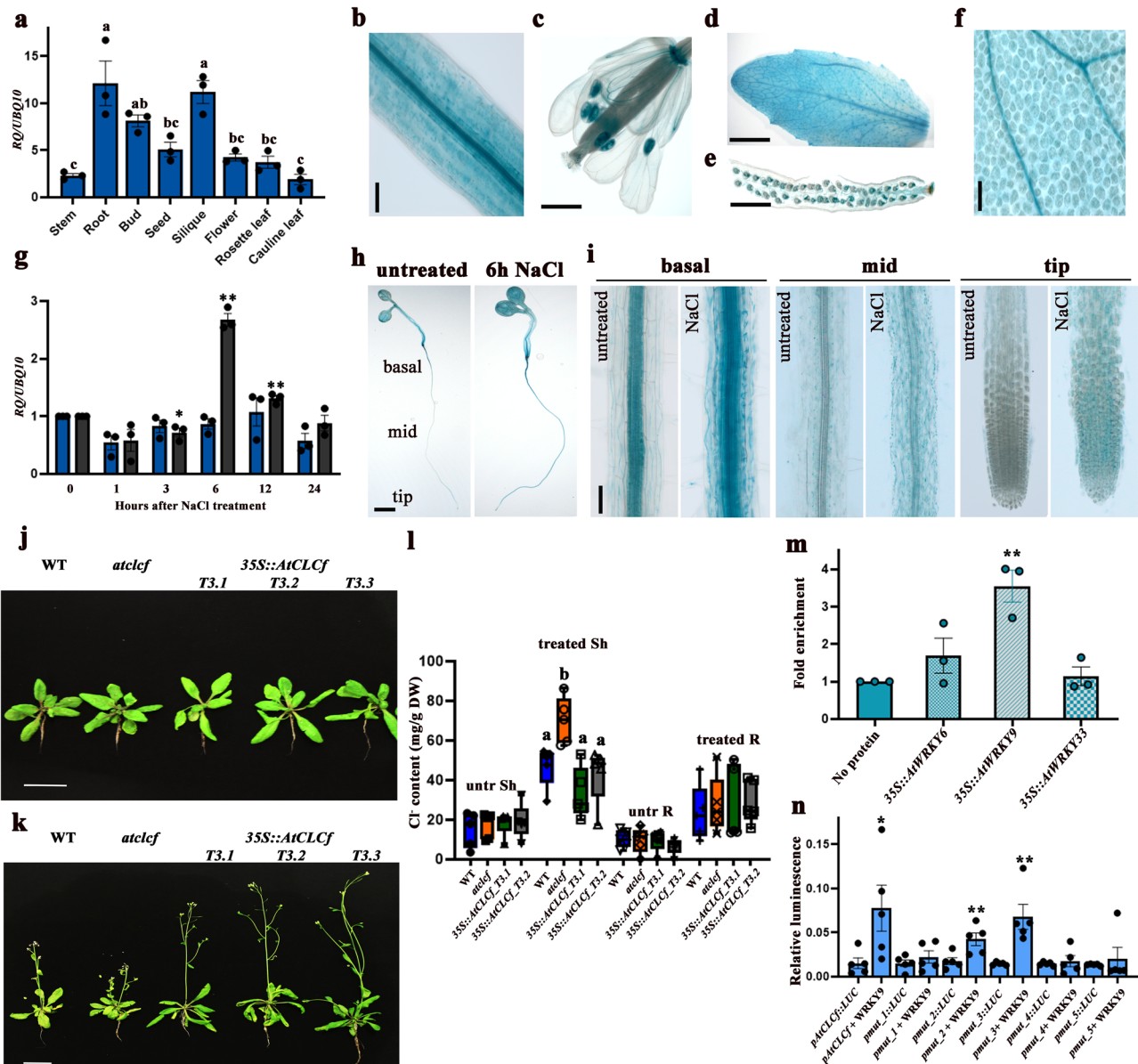

**Fig. 2 | AtCLCf expression is induced by NaCl treatment. a** Tissue-specific expression of *AtCLCf* n = 3 (3 biological replicates, each with 3 technical replicates). Data are mean ± SE, means with different letters within a data set are significantly different, *P* < 0.05 (one-way ANOVA followed by Tukey's test). *pAtCLCf::GUS* expression in one-month-old Arabidopsis (**b**) root, (**c**) flower, (**d**) leaf, (**e**) silique, and (**f**) leaf vasculature. Scale bar in **b** and **f** = 100 μm, **c**–**e** = 500 μm, data were obtained from three independent experiments, and representative images are shown in **b**–**f**. **g** Time-course analysis of *AtCLCf* expression in the shoots and roots of one-week-old Arabidopsis seedlings after 100 mM NaCl treatment. Data are mean ± SE, n = 3 (3 biological replicates, each with 3 technical replicates). Asterisks indicate statistically significant differences (*P < 0.05, **P < 0.01) between 0 h and other time points, as measured by unpaired Student's *t* test (two-tailed). **h** *pAtCLCf::GUS* expression in one-week-old untreated and salt-treated (100 mM NaCl, 6 h) Arabidopsis seedlings. Scale bar = 1 mm, data were obtained from three independent experiments, and representative images are shown here. **i** Close-up images of the roots in **h** Scale bar = 100 μm. **j**, **k** Growth responses of one-month-old WT, *atclcf*, and *35S::AtCLCf* plants grown in soil (**j**) without NaCl treatment and

(**k**) after one week of 150 mM NaCl treatment. Scale bar = 1 cm, (n = 3). **l** Cl⁻ contents in the shoots and roots of WT, *atclcf*, and *35S::AtCLCf* plants under control (untreated) and salt-treated conditions (n = 5). Data are mean ± SE, means with different letters within a data set are significantly different, *P* < 0.05 (one-way ANOVA followed by Tukey's test). The box plots with whiskers show all the data points and their distribution. Horizontal lines within the boxes indicate the median. Minimal and maximal data points are indicated by the lower and the upper whiskers, respectively, untr untreated, sh shoot, R root. **m** ChIP-qPCR analysis shows enrichment of *AtCLCf* promoter fragment in WRKY9 ChIP sample. Data are mean ± SE, n = 3 (3 biological replicates, each with 3 technical replicates). **n** Luciferase assay carried out using the mesophyll protoplasts obtained from leaves of 4-week-old *atwrky9* mutants. *pAtCLCf::LUC* was used as the control and *35S::AtWRKY9* was used as the test. The *AtCLCf* promoter fragments with mutated WRKY binding sites were used as additional controls. Firefly luciferase activity was normalized to Renilla luciferase activity and plotted (Mean ± SE, n = 5). Asterisks in m and n indicate statistically significant differences (*P < 0.05, **P < 0.01) between control and test, as measured by Student's *t* test.

examine the directionality of Cl⁻ transport, we prepared right-side-out (RSO) and inside-out (ISO) PM vesicles (Supplementary Fig. 5) from WT and *atclcf* plants and used them for the fluorescence quenching assay. No significant Cl⁻ transport was observed in the RSO vesicles (Supplementary Fig. 5a). More importantly, WT ISO vesicles showed higher

fluorescence quenching than ISO vesicles from *atclcf* plants (Supplementary Fig. 5b), suggesting that AtCLCf may have a directional Cl⁻ transport activity from within the cell to the outside across the PM. All these results confirm that AtCLCf functions as a Cl⁻/H⁺ antiporter and may be involved in the efflux of Cl⁻ in plant cells.

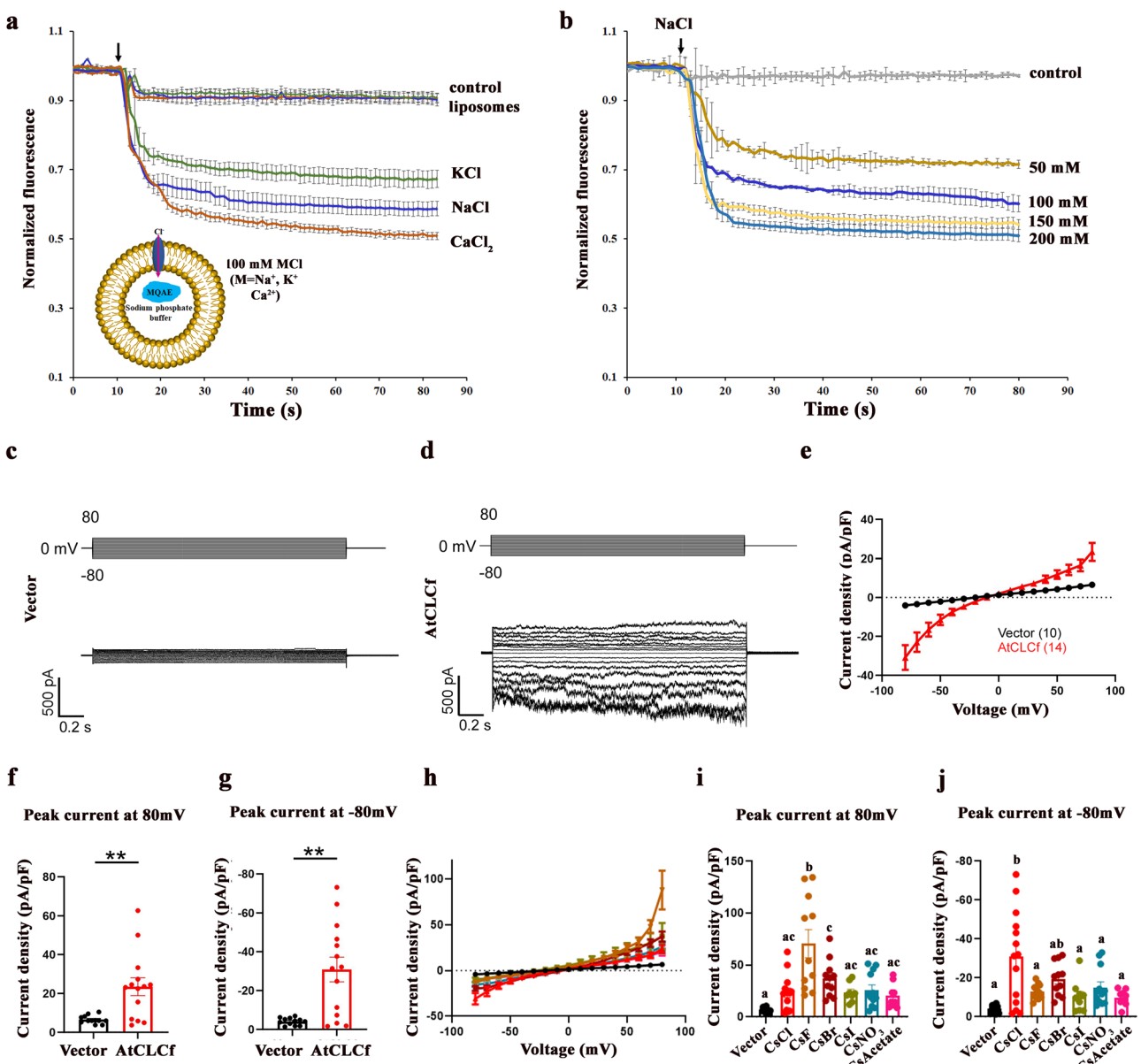

**Fig. 3 | AtCLCf is involved in efflux of Cl⁻. a** Chloride-selective ion transport by MQAE fluorescence quenching assay showed transport of chloride ion by AtCLCf. Data are mean ± SD ($n = 3$). (Inset to **a**) Schematic representation of chloride-selective, MQAE dye-based fluorescence assay for ion transport. Liposomes without incorporation of AtCLCf were used as control. Arrow indicates the time point where different salts were added. The schematic was created using PowerPoint software. **b** AtCLCf showed increased transport of chloride with addition of increasing NaCl concentrations in MQAE based quenching assay. Data are mean ± SD ($n = 3$). **c**–**e** Enhanced inward and outward current was observed in cells expressing AtCLCf as compared to the vector transfected cells. **c** Exemplar traces of a vector trans-fected cell showing the current evoked from holding potential of 0 mV to different test potential ranging from −80 mV to 80 mV with 10 mV increment. **d** Exemplar trace of AtCLCf cell. **e** Current voltage relationship of respective groups. **f**, **g** Data

comparing the maximal outward and inward current at 80 and −80 mV respec-tively. **$P < 0.01$ (**e**–**g** data are mean ± SE, vector $n = 12$, AtClCf $n = 14$). Statistics was performed with unpaired Student's $t$ test (2-tailed). **h**–**j** The effect of replacing most of internal CsCl with CsF, CsBr, CsI, CsNO₃ or CsAcetate on the ionic current. **h** Current voltage relationship of AtCLCf transfected cell recorded with 140 mM CsCl-based internal as compared to cell recorded with 130 mM CsF, CsBr, CsI, CsNO₃ or CsAcetate and 10 mM CsCl-based internal solution. The profiles for vector and AtCLCf cell recorded with CsCl based internal solution were reproduced from **e** for comparison, **i** and **j** data comparing the maximal outward and inward current at 80 and −80 mV respectively. (**h**–**j** data are mean ± SE, $n = 12$ for CsF, 11 for CsBr, 9 for CsI, 12 for CsNO₃ and 9 for CsAcetate). Different letters within a data set are sig-nificantly different, $P < 0.01$ (one-way ANOVA followed by Tukey's test).

To gain structural insights into the roles of specific amino acid residues in AtCLCf function, we generated a homology model (Fig. 4a–c) based on the crystal structure of *Salmonella typhimurium* CLC[9], which shares 48% sequence similarity with AtCLCf. Three selec-tivity filter domains were identified in AtCLCf. We mutated five con-served amino acid residues in these domains and studied their effects on AtCLCf function using yeast growth assays (Fig. 4b–d). Specifically, we assayed the Cl⁻ transport function of AtCLCf by introducing the WT

*AtCLCf* gene and *AtCLCf* genes with mutations in the three selectivity filter domains (domain 1, *Δgef1_MD1*; domain 2, *Δgef1_MD2*; and domain 3, *Δgef1_MD3*) into the yeast (*Saccharomyces cerevisiae*) deletion mutant *Δgef1*, which lacks the *GEF1* gene (a yeast homolog of *CLC* genes). The salt sensitivity of *Δgef1* mutant strain was rescued by the introduced WT *AtCLCf* gene and MD1, while introduction of the other two domain mutants (*MD2* and *MD3*) failed to rescue the salt sensi-tivity (Fig. 4e).

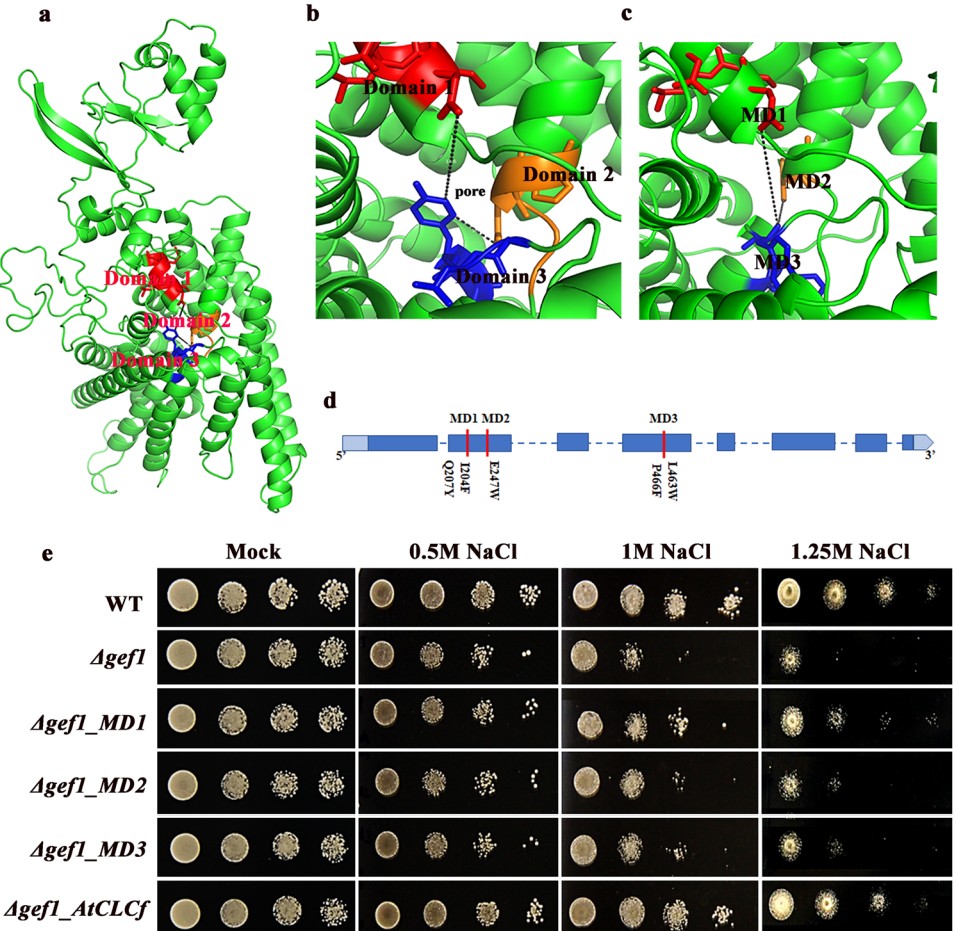

**Fig. 4 | Structural and functional analysis of AtCLCf. a** Homology model of AtCLCf based on the crystal structure of *Salmonella typhimurium* CLC was generated using PyMOL software and it shows three selectivity filter domains. **b** Enlarged view of (**a**) showing the selectivity filter domains and the pore. **c, d** Positions of mutations in the selectivity filter domains: MD1(I204F, Q207Y); MD2(E247W); MD3(L463W, P466F). **e** Growth of yeast strain BY4741 and its derivative *Δgef1* harboring mutated domains (*MD1, MD2,* and *MD3*) and *AtCLCf* in the YEp352 plasmid grown on SD/-Ura medium with (0.5–1.25 M) and without NaCl, data were obtained from three independent experiments, and representative images are shown here.

To confirm the Cl⁻ transport function of AtCLCf in plant cells, we introduced the constructs harboring mutated *AtCLCf* and *35S::AtCLCf* into *atclcf* plants and measured the root length of the transgenic seedlings. There was no significant difference in root length between these genotypes in the absence of salt treatment (Fig. 5a). Upon NaCl treatment, *35S::AtCLCf;atclcf* plants exhibited higher salt tolerance than *atclcf* plants, with their root length restored to WT levels. By contrast, transgenic plants harboring mutated *35S::MD2;atclcf* (E247W) and *35S::MD3;atclcf* (L463W and P466F) behaved similar to *atclcf* seedlings, indicating that the expression of these constructs failed to rescue the increased salt sensitivity of the mutant (Fig. 5b, c). These results show that the residue that is altered in the *35S::MD2;atclcf* mutant, namely, glutamate 247 in the selectivity filter domain 2, is important for the functioning of AtCLCf, as has also been found in other organisms[9,10].

Having demonstrated that AtCLCf is a functional Cl⁻/H⁺ antiporter containing the typical glutamate in its selectivity filter domain, we performed protoplast-based assays to attempt to verify the directionality of transport in the plant. An MQAE assay using mesophyll protoplasts showed that more Cl⁻ was retained in *atclcf* and *35S::MD2;atclcf* protoplasts (exhibit low fluorescence = more quenching) than in *35S::AtCLCf;atclcf* and WT protoplasts (Fig. 5d, e). In an independent assay, we quantified the fluorescence quenching (via spectrofluorimetry) of salt-treated protoplasts loaded with MQAE. Similar to the above observations (microscopy), the *atclcf* and *35S::MD2;atclcf* protoplasts showed significantly higher quenching,

which shows that more Cl⁻ is retained in them as a result of reduced Cl⁻ efflux compared to the *35S::AtCLCf;atclcf* and WT protoplasts (Supplementary Fig. 7). Additionally, there was a significant reduction in Cl⁻ efflux from *atclcf* seedling roots compared to those of the WT and *35S::AtCLCf;atclcf* (Fig. 5f). Collectively, these findings show that AtCLCf is important for removal of excess Cl⁻ from the roots.

## Salt stress triggers AtCLCf translocation from the Golgi apparatus to the PM

The earlier observation that AtCLCf localizes to the Golgi apparatus or TGN/EE[15,16,21] does not adequately explain how this protein confers salt tolerance at either the cellular or plant level. To address this issue, we first examined the protein's subcellular localization by transiently expressing *35S::GFP-AtCLCf* in *Nicotiana benthamiana* leaf epidermal cells. In untreated cells, GFP-tagged AtCLCf localized to the Golgi along with the Golgi marker G-rk CD3-967 (Fig. 6a). Interestingly, after 4 h of NaCl treatment, GFP-AtCLCf progressively spread via the TGN (along with the TGN marker SYP61-RFP) to the PM (Fig. 6b). After 6 h of NaCl treatment, the GFP-AtCLCf signal was mostly concentrated in the PM (Fig. 6c and Supplementary Video 1) along with the PM marker (PM-rk CD3-1007). There was about 60% increase in the translocation of AtCLCf to the PM compared to the 0 h value (Fig. 6d), indicating that NaCl induced this subcellular translocation. A similar stimulatory effect of NaCl on the formation of TGVs and their fusion with vacuoles/PM was previously demonstrated for FLOTILLIN1 in Arabidopsis[22]. We

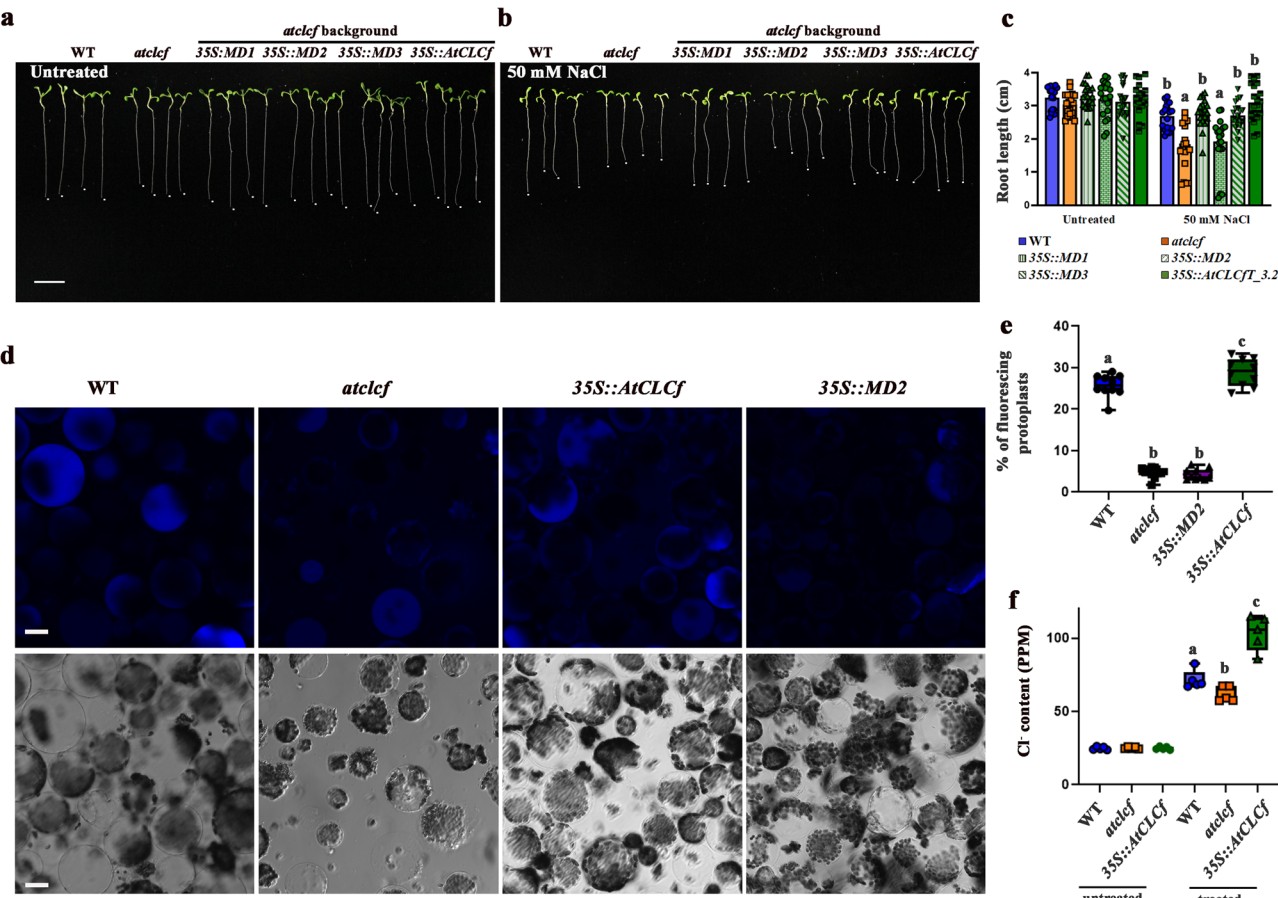

**Fig. 5 | The glutamate (E247) residue in selectivity filter domain 2 is important for AtCLCf function.** Root lengths of WT, *atclcf*, *35S::AtCLCf;atclcf*, and *35S::MD1-MD3AtCLCf;atclcf* seedlings, under (**a**) untreated and (**b**) treated with 50 mM NaCl conditions; scale bar, 1 cm. Root tips are marked by white dots. **c** Root lengths of seedlings of the indicated genotypes with (50 mM) and without NaCl treatment. Root lengths were measured after one week of germination, Mean ± SE, *n* = 20. **d** Results of Cl⁻ transport assay using mesophyll protoplasts of one-month-old WT, *atclcf*, *35S::AtCLCf;atclcf*, and *35S::MD2AtCLCf;atclcf* (*35S::MD2*) plants. Protoplasts treated with 50 mM NaCl for 3 h were loaded with 100 μM MQAE for 30 min and visualized under a confocal microscope. Lower panel shows the respective brightfield images; scale bars, 20 μm. **e** Quantification of the data in **d**. Mean ± SE, *n* = 10. **f** Cl⁻ efflux assay (as described in the methods) was carried out from 10-day-old seedlings of WT, *atclcf* and *35S::AtCLCf;atclcf* (*n* = 5). Means with different letters within a data set are significantly different, *P* < 0.05 (one-way ANOVA followed by Tukey's test). Data were obtained from three independent experiments, and representative images are shown here. Horizontal lines within the boxes in **e** and **f** indicate the median. Minimal and maximal data points are indicated by the lower and the upper whiskers, respectively.

further confirmed the NaCl-induced translocation of AtCLCf to the PM in Arabidopsis leaf protoplasts expressing AtCLCf under its native promoter (*pAtCLCf::GFP-AtCLCf*) (Fig. 6e). These data suggest that the underlying translocation mechanism involves the exocytic trafficking pathway. We also detected AtCLCf expression in the root cortex and epidermis of *pAtCLCf::GFP-AtCLCf* seedlings within 6 h of salt treatment (Supplementary Fig. 8a, b). Notably, the differences in the root elongation responses of the lines were observed within the same timeframe of a 6 h NaCl treatment (Fig. 1a, b). Thus, we identified a salt-induced translocation of AtCLCf to the PM, providing a rapid mechanism to help Arabidopsis root cells adapt to salt stress.

### AtRABA1b/BEX5 mediates the salt-induced translocation of AtCLCf from the Golgi to the PM

Similar to what we observed in *N. benthamiana* and Arabidopsis (leaves and protoplasts), GFP-AtCLCf translocated to the PM upon NaCl treatment in *35S::GFP-AtCLCf* roots (Fig. 7a). Hence, we used this system to test if the translocation mechanism indeed involves the exocytic trafficking pathway. After 3 h of treatment with the trafficking inhibitor brefeldin-A (BFA) alone or with BFA plus 50 mM NaCl, GFP-AtCLCf proteins aggregated with the endosomal membrane to form BFA bodies and failed to be translocated to the PM (Fig. 7a, b), which is consistent with the known effect of BFA on exocytic trafficking[23,24]. These results indicate that BFA prevents the translocation of AtCLCf to the PM even under salt stress. Furthermore, root elongation was reduced in seedlings treated with 5 μM BFA plus 50 mM NaCl (Fig. 7c), mimicking the phenotype of *atclcf* roots.

Recycling between the PM and endosomes is a characteristic of plant PM proteins[23–26]. We hypothesized that Rab GTPases, which are crucial regulators of vesicle trafficking[27,28] might be involved in the translocation of AtCLCf to the PM. We focused on AtRABA1b (also called BEX5) as the potential facilitator because it was reported to be involved in the transport of cargos between the TGN and PM, as well as plant salt tolerance[23,29,30]. Indeed, when expressed in *bex5* protoplasts, RFP-AtCLCf failed to be translocated to the PM even under NaCl treatment (Fig. 7d and Supplementary Fig. 8c), and *bex5* seedlings were sensitive to BFA treatment (Fig. 7c). To further confirm that AtRABA1b is involved in the trafficking of AtCLCf, we used two AtRABA1b mutant proteins: the GTP-bound form, GFP-AtRABA1b^Q72L (QL), and the GDP-bound form, GFP-AtRABA1b^S27N (SN). While QL acts as a constitutively active RAB GTPase and accumulates on membranes, SN is a dominant negative form whose accumulation on membranes is significantly lower than that of WT AtRABA1b[23,29]. When RFP-AtCLCf was expressed in the leaf protoplasts of plants harboring QL (QL plants), significantly more AtCLCf localized to the PM than in WT protoplasts. By contrast,

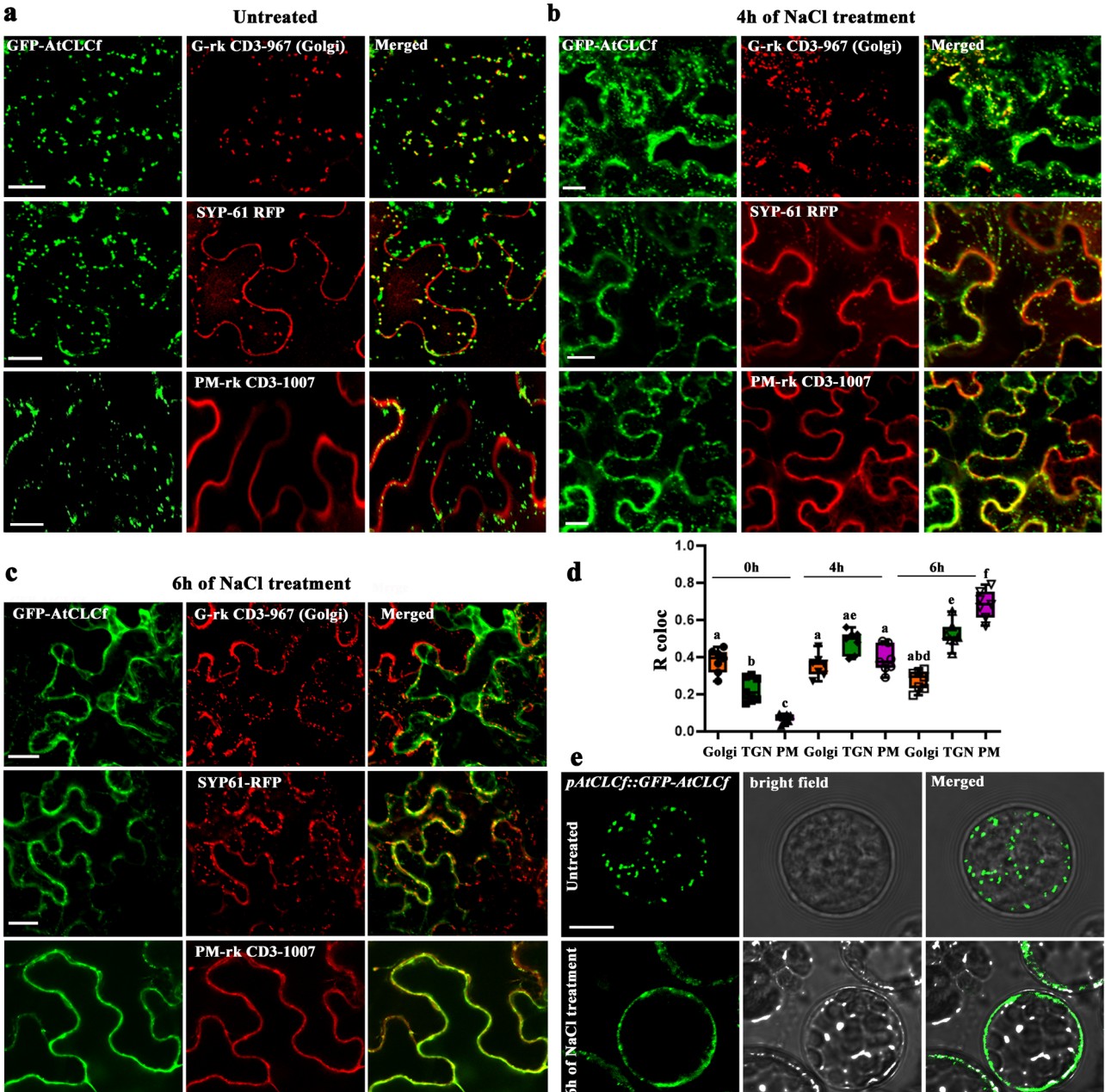

**Fig. 6 | NaCl induces the translocation of AtCLCf from the Golgi to the PM.**
**a**–**c** Confocal microscopy images of *Nicotiana benthamiana* leaf epidermal cells show that N-terminal GFP-tagged AtCLCf translocates to the PM when treated with 100 mM NaCl, as indicated by the expression of GFP-AtCLCf along with Golgi, TGN, and PM markers. Shown are GFP-AtCLCf expression (**a**) before and after (**b**) 4 h and (**c**) 6 h of NaCl treatment. The GFP signal was visualized under 488 nm excitation, 500–525 nm emission; the markers were visualized under 561 nm excitation, 575–600 nm emission. Co-localization of GFP-AtCLCf with markers is shown in the merged images. Scale bars, 20 μm, data were obtained from three independent experiments, and representative images are shown here. **d** Quantification of AtCLCf-GFP localization with respective markers (*n* = 7) using ImageJ. Horizontal lines within the boxes in d indicate the median. Minimal and maximal data points are indicated by the lower and the upper whiskers, respectively. **e** Expression of GFP-tagged AtCLCf in the leaf mesophyll protoplasts of *pAtCLCf::GFP-AtCLCf* Arabidopsis lines without and with NaCl treatment, *n* = 10. Scale bars, 10 μm. Means with different letters are significantly different, *P* < 0.05 (one-way ANOVA followed by Tukey's test).

the localization pattern of AtCLCf in SN protoplasts was similar to that in *bex5* protoplasts (Fig. 7d and Supplementary Fig. 8c). Finally, when seedlings were grown in the presence of 50 mM NaCl, the root lengths of the *atclcf*, *bex5* and SN genotypes were significantly reduced (~40%) compared to those of untreated controls (Fig. 7e, f), whereas this treatment had less of an effect on WT, *35S::AtCLCf;atclcf* and QL seedlings. A similar effect of salt was seen on the seedling root growth of SN and QL plants in an earlier study[29]. Collectively, these findings indicate that AtRABA1b/BEX5 facilitates the NaCl-induced translocation of AtCLCf to the PM to enhance plant salinity tolerance (Fig. 7g). This represents a unique mechanism of salt adaptation.

## Discussion

Studies on the effect of salinity stress on plants have mostly focused on the involvement of Na⁺, while the counter-anions such as Cl⁻ are often neglected. Hence, little is known about the molecular and cellular mechanisms of plant tolerance to Cl⁻ ion toxicity imposed by salinity stress[6]. Plants respond to salinity by sequestration or efflux of ions

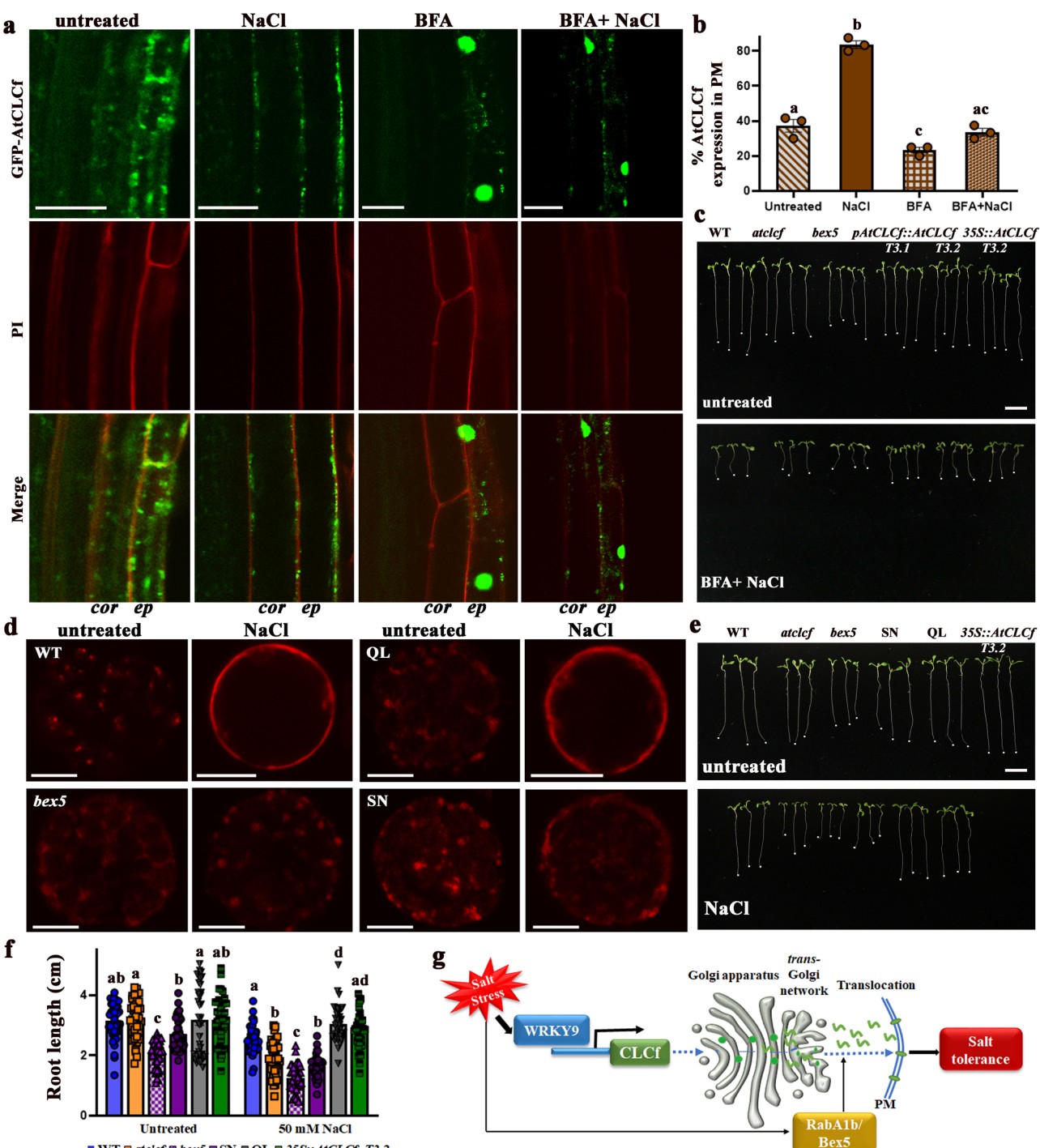

**Fig. 7 | NaCl-induced translocation of AtCLCf from the Golgi to the PM is mediated by AtRABA1b/BEX5. a** Expression of GFP-AtCLCf in the roots of one-week-old Arabidopsis seedlings under various treatments (untreated; 50 mM NaCl; 50 μM brefeldin-A (BFA) for 3 h; 50 mM NaCl +50 μM BFA for 3 h). Scale bars, 20 μm; cor cortex, ep epidermis. **b** Quantification of GFP-AtCLCf expression in the PM and BFA bodies (mean ± SE, *n* = 15). Means with different letters within a data set are significantly different, *P* < 0.05. (one-way ANOVA followed by Tukey's test). **c** Growth of one-week-old seedlings treated with 0 and 5 μM BFA + 50 mM NaCl. **d** Representative confocal images showing RFP-AtCLCf localization in untreated and 50 mM NaCl-treated WT, *bex5*, QL, and SN leaf protoplasts

(561 nm excitation, 575–600 nm emission); scale bars, 10 μm. **e, f** Root lengths of one-week-old seedlings [WT, *atclcf*, *bex5*, SN, QL, and *35S::AtCLCf;atclcf* (left to right) lines without (untreated) and with 50 mM NaCl]. Scale bar, 1 cm. Data are mean ± SE, *n* = 40. Means with different letters within a data set are significantly different, *P* < 0.05. (one-way ANOVA followed by Tukey's test). **g** Model shows that the NaCl-induced translocation of AtCLCf from the Golgi to the PM to enhance salinity tolerance is mediated by AtRABA1b/BEX5. Data in **a** and **d** were obtained from three independent experiments, and representative images are shown here. Root tips are marked by white dots in **c** and **e**.

specifically to reduce cytosolic Na$^+$ and Cl$^-$ concentrations[31,32]. Several vacuolar sodium hydrogen exchangers sequester Na$^+$ into the vacuole under salt stress[33], but only Salt Overly Sensitive 1, a PM Na$^+$/H$^+$ antiporter, has been extensively characterized as a Na$^+$ efflux transporter in plants[34–36].

Accumulation of excessive Cl$^-$ reduces the growth of plants by decreasing the photosynthesis rate and leaf water potential, increasing ROS production and causing stomatal closure[5,31]. The effect of salt stress due to Cl$^-$ toxicity can be mitigated by means of Cl$^-$ efflux to maintain intracellular Cl$^-$ homeostasis particularly in salt-sensitive plants such as glycophytes[37,38]. To reduce Cl$^-$ toxicity in plants, ion channels such as members of the CLC family are predicted to be involved. The expression of GmCLC1 helped to prevent damage to cytoplasmic and organelle membranes caused by excess Cl$^-$[39]. Plant CLC proteins play a major role in balancing the anion concentration under salinity. Thus, ectopic expression of some Cl$^-$ channels in targeted tissues or in whole plants have been reported to increase their salinity tolerance, but the mechanism of action is poorly understood[8,40]. In Arabidopsis, seven genes (*AtCLCa-g*) encoding CLC channel/transporter proteins have been identified. AtCLCc and AtCLCg are located in the tonoplast and are known to improve salt tolerance of plants[41,42]. In our study, AtCLCf enhanced the salt tolerance of both seedlings and mature plants (Figs. 1 and 2), which is in agreement with the finding that other plant Cl$^-$ transporters affect salt tolerance[43–45].

There are no prior reports on the molecular regulation of CLC family genes thus far. Significant suppression of *AtCLCf* in *atwrky9* mutant, and the observed interaction of *AtCLCf* promoter with AtWRKY9 suggest that AtWRKY9 is a key regulator of *AtCLCf* expression as part of the salt stress signal transduction mechanism. This conclusion is also supported by the multiple reports on the involvement of various *WRKY* genes in salt and drought tolerance of plants[19,46,47].

It was evident from the homology model generated (Fig. 4a–c) based on the *Salmonella typhimurium* crystal structure[48] that the conserved Cl$^-$ selectivity filter domain is also present in AtCLCf. To functionally characterize plant ion transporters, heterologous expression of Cl$^-$ and Na$^+$ transporters in yeast mutants has been extensively used[35,39,44,45]. *Δgef1* yeast mutant was used to understand the role of CLC transporters in regulating the ion homeostasis under salt/Cl$^-$ stress. In our study, AtCLCf was able to complement the salt sensitive phenotype of *Δgef1* mutant. When some of the critical amino acid residues in two of the three selectivity filter domains were substituted [MD2: (E247W), MD3: (L463W, P466F)], their ability to complement *Δgef1* was lost (Fig. 4e). Parallel to the yeast results, *atclcf* lines with mutation in these specific domains showed higher sensitivity towards salt and accumulated more Cl$^-$ in the protoplasts compared to *AtCLCf* ectopic expression lines, which indicates that these conserved domains play an important role in the protein's function as a chloride channel. These results also imply that introducing these amino acid substitutions influences the function of AtCLCf by affecting its conformation/transport activity. Moreover, the residue that is altered in the *35S::MD2;atclcf* mutant, glutamate 247 in the selectivity filter domain 2, is important for the functioning of AtCLCf, as has also been found in other organisms[9,10]. The observation that AtCLCf shows transport activity of H$^+$ ions in both electrophysiological experiments and HPTS assays (Supplementary Fig. 6) suggests that it acts as a Cl$^-$/H$^+$ antiporter with higher specificity for Cl$^-$ transport. While some of the CLCs are reported to show NO$_3$ transport activity, our results show that AtCLCf is not involved in NO$_3$ transport[14,15,49].

Previous studies[21] had indicated that AtCLCf is present within the Golgi and TGN/EE[15,16,21]. Our initial studies with transient assay in *N. benthamiana* helped to confirm this. Based on such information of subcellular localization of channels/transporters in the endosomal membranes, such as AtCLCd, AtCLCf, AtNHX5, and AtNHX6, it was interpreted that they might directly contribute to Cl$^-$ and Na$^+$ homeostasis in the endomembrane space of plant cells under salt stress conditions[50]. However, our study revealed a novel mode of action of AtCLCf that involves a salt-mediated translocation of the protein from the Golgi to PM via TGN (Fig. 6). TGN is a cellular compartment where endocytic and secretory pathways intersect, and it mediates the recycling of endocytic vesicles back to the PM. This helps to explain why *atclcf* seedlings showed decreased Cl$^-$ efflux compared to the *35S::AtCLCf;atclcf* seedlings upon salt treatment (Fig. 5). Additionally, the observed increase of *AtCLCf* transcripts in the roots and AtCLCf expression in the Arabidopsis root cortex/epidermis after salt treatment (Supplementary Fig. 8) further shows that AtCLCf might modulate Cl$^-$ efflux from roots under salt stress. This is supported by the observation that *atclcf* ISO vesicles failed to show any transport activity (Supplementary Fig. 5), along with similar data from the quantitative fluorescence assays with protoplasts (Supplementary Fig. 7). Our electrophysiological analyses in HEK293FT cells further confirm our findings that the AtCLCf is involved in Cl$^-$ efflux (Fig. 3). This efflux mechanism could prevent the accumulation of Cl$^-$ in shoots leading to increased salt tolerance observed in the ectopic expression lines (Figs. 1 and 2). The fact that BFA, a known inhibitor for protein secretion and membrane trafficking[51] effectively blocked the PM translocation of AtCLCf, which partially explains why all the tested Arabidopsis genotypes showed short root phenotype upon BFA treatment. The possibility of BSA blocking other trafficking pathways such as that of auxin[52] in addition to the AtCLCf translocation cannot be dismissed (Fig. 7). Also, the protoplasts from AtRABA1b$^{Q72L}$ plants showed increased expression of AtCLCf on the PM, further confirming that the translocation occurs through the vesicular trafficking process involving AtRABA1b/BEX5. This represents the cellular/molecular mechanism to alleviate the Cl$^-$ ion toxicity under salt stress. The finding that a chloride channel is translocated to the PM in plants has wider significance and needs to be understood in great detail because in humans, mutations in Barttin, an accessory protein required for the functioning and trafficking of a human chloride channel (CIC-K) to the PM, lead to symptoms such as renal salt loss and deafness[12].

Overall, our study revealed how NaCl modulates AtCLCf activity by triggering its translocation from the Golgi apparatus to the PM to increase the Cl$^-$ efflux capacity in roots, representing an essential aspect of salt tolerance in plants. This knowledge could be used to improve salinity tolerance in crops in the future.

## Methods

### Plant materials and growth conditions

T-DNA insertional mutants, *atclcf1* (SAIL_174_G06), *atclcf2* (SALK_059540.1), and wild-type (WT) *Arabidopsis thaliana*, ecotype Columbia-0 were purchased from the Arabidopsis Biological Resource Centre (ABRC). The overexpression line, *35S::AtCLCf* was generated in WT background, and the complementation line *pAtCLCf::AtCLCf;atclcf* was generated in *atclcf* background by us. The SN and QL lines were gifts from Dr. Tomohiro Uemura. Cold stratified Arabidopsis seeds were grown on MS medium solidified with Gelrite (0.5% w/v), with or without treatments (50 mM NaCl/KCl/NaNO$_3$/mannitol or 5 μM BFA or 5 μM BFA + 50 mM NaCl) for 1 week and used for the following analyses: a) Root length assays from untreated and treated one-week-old seedlings were performed using Image J software[53] and root length in cm was plotted as mean ± SE, $n \geq 15$. b). Root fine phenotyping was carried out by treating the one-week-old seedlings with 50 mM NaCl for 6 h by submerging the seedlings with the salt solution while the seedlings were growing on the semi-solid MS plates. Following the treatment, seedlings were fixed in 1% PFA and imaged using a LEICA CTR5000 differential interference contrast (DIC) microscope equipped with a Nikon DS-Ri1 camera. Effect of NaCl treatment on meristematic zone (MZ, quiescent centre to the first elongated cortical cell), and elongation zone (EZ, first elongated cortical cell to the appearance of

bulging of root hairs from the trichoblasts) were analyzed using Image J software and the length of each zone was plotted as mean ± SE, $n = 10$. For studies with older seedlings, one-month-old seedlings were treated with 150 mM NaCl for one week. Chlorophyll content and FW measurements were carried out as described earlier[54]. For survival analysis, the pots were flushed with water twice to remove the soil-bound NaCl followed by a recovery growth in NaCl-free water for one week.

## In silico analysis

The NCBI database was used as a search engine for nucleotide and protein sequences. Expasy tool (https://web.expasy.org/translate/) was used to translate the CDS sequences to amino acid sequences and multAlin (http://multalin.toulouse.inra.fr/multalin/) was used to align the amino acid sequences. Phylogenetic analysis was carried out using http://www.phylogeny.fr/. Primers for qRT-PCR were designed using NCBI. Transmembrane helices of the proteins were predicted using TMHMM.

## Cloning and generation of transgenic Arabidopsis lines

Mutant lines *atclcf1* (SAIL_174_G06) *and atclcf2* (SALK_059540.1*)* with T-DNA insertions were obtained from the SALK collection[55]. Positions of T-DNA insertion sites are shown in Supplementary Fig. 9a, b. Plants homozygous for the T-DNA insertion were selected by genotyping with primers designed using the T-DNA primer design tool (http://signal.salk.edu/tdnaprimers.2.html). Seeds were collected only from those *atclcf* lines that showed more than 80% suppression of *AtCLCf* in qRT-PCR analysis (Supplementary Fig. 9c). For generation of overexpression lines in Arabidopsis, coding sequence of *AtCLCf* was cloned into *pGREEN* binary vector. The constructs *35S::AtCLCf, 35S::GFP-AtCLCf, pAtCLCf::GUS* and *pAtCLCf::AtCLCf* were individually electroporated into *Agrobacterium tumefaciens* strain GV3101:pMP90 and introduced into WT or *atclcf* plants by the floral dip method[56]. Basta-resistant T1 transgenic plants were selected after confirmation by genotyping PCR and qRT-PCR (Supplementary Fig. 9d) analyses. For chromatin immunoprecipitation (ChIP) assay, CDS of *AtWRKY9, AtWRKY6* and *AtWRKY33* were cloned into pGreen binary vector with hemagglutinin (HA) fusion tag. All the plasmids were sequence verified before use and the primers used in the study are listed in Supplementary Table 2.

## Chromatin immunoprecipitation (ChIP) using Arabidopsis protoplasts

Mesophyll protoplasts were isolated from leaves of 3- to 4-week-old WT Arabidopsis (Col-0) plants and transfected as described earlier[57] with minor modifications. For each transfection, 10 μg of purified plasmid DNA (*35S::AtWRKY6, 35S::AtWRKY9* or *35S::AtWRKY33*) was used. Polyethylene glycol (PEG)-CaCl$_2$ transfection solution used was as follows: 25 % PEG, 0.4 M mannitol, and 150 mM CaCl$_2$. The transfected protoplasts were incubated for 20 h at room temperature and fixed with formaldehyde. Protoplasts with empty plasmids were used as the negative control. ChIP was performed as described previously[58], with minor changes. Anti-HA monoclonal antibody (Santa Cruz Biotechnology) bound to Protein-A agarose beads (Sigma) was used to immunoprecipitate the genomic DNA fragments. ChIP-qPCR analysis was carried out as described[59] to check for enrichment of promoter fragment in the final eluted chromatin. Fold change (promoter fragment vs. no protein control) was plotted using data from 3 independent biological replicates each with three technical replicates.

## Luciferase assay using Arabidopsis protoplasts

Mesophyll protoplasts were isolated from leaves of 4-week-old *atwrky9* mutant seedlings, and the luciferase assay was carried out as described earlier[60]. 2 kb upstream sequence of *AtCLCf* was cloned into the *pGreen II-0800-LUC* vector to generate the reporter. The vector with the *pAtCLCf* promoter (*pAtCLCf*::LUC) was used as a reference

control while *35S::AtWRKY9* was used as the effector. Five WRKY binding sites in the *AtCLCf* promoter fragment were mutated (TTGAC to TTacC) by site directed mutagenesis, and the mutant promoter was cloned into *pGreen II-0800-LUC* vector and used as an additional control. The luciferase assay was carried out using the Dual-Luciferase® Reporter Assay System (Promega) following the manufacturer's instructions. The luminescence was measured using the GloMax discover (Promega). Firefly luciferase activity was normalized to Renilla luciferase activity. Data shown were taken from five independent biological replicates each with three technical replicates.

## Subcellular localization of AtCLCf in *N. benthamiana* leaf epidermal cells

The coding sequence of GFP was fused in-frame at the N-terminal of *AtCLCf* in *pGREEN* plasmid. Empty vector, *35S::GFP-pGREEN* was used as the control. These plasmids as well as the plasmids containing subcellular markers (Golgi, TGN and PM) were introduced into *Agrobacterium*. The subcellular marker plasmids (Golgi-rk CD3-967 and PM-rk CD3-1007) were obtained from TAIR. SYP61-RFP[61] was a gift from Dr. Shen Quan Pan. Mature leaves from 3- to 4-week-old *N. benthamiana* plants were co-infiltrated with *Agrobacterium* harboring *35S::GFP-AtCLCf* and the related subcellular marker constructs. *N. benthamiana* leaf epidermal cells were examined for GFP-AtCLCf expression along with mCherry-tagged markers using a confocal laser scanning microscope (Olympus FV3000) with 488 nm excitation, 500–525 nm emission: and 561 nm excitation, 575–600 nm emission, respectively. For salt treatment, *N. benthamiana* leaves were infiltrated with 100 mM NaCl and imaged after 4 and 6 h. ImageJ software was used to measure the colocalization of AtCLCf with Golgi, TGN and PM markers and plotted as Rcoloc values. The Rcoloc values were calculated using Returns Pearson's correlation coefficient for pixels where both GFP and RFP are above their respective threshold. This generates a value for each channel. For Ch1, this value is equal to the sum of the pixel intensities with intensities above both Ch1 and Ch2 thresholds i.e. (sum of Ch1 pixel intensities in the Yellow area) ÷ (sum of Ch1 pixels intensities in the Red+Green+Yellow areas).

## Subcellular localization of AtCLCf in Arabidopsis leaf protoplasts

The coding sequence of GFP or RFP were fused in-frame at the N-terminal of *AtCLCf* in *pGREEN* plasmid. Mesophyll leaf protoplasts were isolated from one-month-old plants of WT, *bex5*, SN and QL genotypes as described earlier[57]. The protoplasts were transfected with 10 μg of *35S::GFP-AtCLCf* or *35S::RFP-AtCLCf* using a polyethylene glycol-calcium chloride transfection solution as described previously[54]. After the transfection for 16 h, untreated and treated (50 mM NaCl for 3 h) protoplasts were visualized under confocal laser scanning microscope (Olympus FV3000). RFP-AtCLCf was visualized using 561 nm excitation and 600–620 nm emission. For each measurement of AtCLCf localization in PM, ~20 protoplasts from each genotype were used and the data are represented as mean ± SE.

## Homology modeling of AtCLCf and site directed mutagenesis of the selectivity filter domain residues

A homology model of AtCLCf was constructed based on the crystal structure of *Salmonella typhimurium* CLC[9], using the PyMOL software (https://pymol.org/2/). Pore lining residues present in the selectivity filter domains 1, 2 and 3 were mutated to study their role on Cl⁻ transport function of AtCLCf. The following residues were mutated in domains 1–3 and labeled as mutated domain (MD) 1–3; MD1 (Isoleucine (I) 204 to Phenylalanine (F), Glutamine (Q) 207 to Tyrosine (Y)), MD2 (Glutamate (E) 247 to Tryptophan (W) and MD3 (Lysine (L) 463 to W, Proline (P) 466 to F). PCR-based site directed mutagenesis was carried out as described earlier[62], followed by cloning into *pGREEN* binary vector. Using these mutant *AtCLCf-pGREEN* constructs,

*35S::AtCLCfMD1*, *35SS::AtCLCfMD2* and *35S::AtCLCfMD3* lines were generated in the *atclcf* background to study their response to NaCl treatment and Cl⁻ transport functionality.

## PM isolation by two-phase partitioning and formation of inside-out vesicles

The detailed methods are given in Supplementary Information. Briefly, plasma membranes (predominantly right-side-out vesicles) were purified from a microsomal fraction of 6-week-old Arabidopsis seedlings (150 g FW) by partitioning in an aqueous polymer two-phase system as described earlier[63]. Pelleted PM proteins were resuspended in PM washing buffer complemented with protease inhibitor and 5 mM DTT, frozen in liquid nitrogen, and stored for later use at −80 °C. This mainly consisted of the right-side-out vesicles. Using these highly purified right-side-out PM vesicles inside-out vesicles were prepared as described earlier[64].

## Spectrofluorimetric analysis of Cl⁻ transport in PM vesicles

For spectrofluorimetric analysis, N-(ethoxycarbonylmethyl)-6-methoxyquinolium bromide (MQAE)-loaded ISO and RSO PM vesicles were prepared by incubation of vesicles in a hypotonic medium for 15 min at 37 °C. The hypotonic medium comprised of Hanks' balanced salt solution (HBSS) diluted 1:1 with water and contained 5 mM MQAE. At the end of the loading period, 100 μl samples of cell suspension were diluted 15:1 in HBSS, centrifuged, resuspended in 200 μl of HBSS and then incubated for a further 15 min at 37 °C to allow recovery from the hypotonic shock. The spectrofluorimetric analysis was performed with these vesicles, stirred magnetically with a glass fly inside a standard cm path-length quartz fluorescence cuvette. A constant temperature of 20 °C was maintained by water circulating through the cuvette holder of the spectrofluorometer. Reaction was started by adding 2 ml of suspension medium (50 μl ISO/RSO MQAE-loaded PM vesicles in 5 mM phosphate buffer) was added to the cuvette. After one minute, 50 mM NaCl was added to the buffer to determine the fluorescence response. The excitation and emission wavelengths used for MQAE were 360 nm and 450 nm, respectively. The fluorescence values were normalized to 1 and plotted to show the quenching of fluorescence after addition of NaCl.

## Liposome-based spectrofluorimetric analysis of Cl⁻ transport

Codon optimization and recombinant protein expression: In order to carry out liposome-based assays, synthetic gene expression construct was prepared by cloning the codon-optimized (for yeast, *Pichia pastoris*), full length *AtCLCf* gene into yeast expression vector pPICZA (GenScript Biotech, Singapore) that was used for recombinant protein production subsequently. The original and codon-optimized sequence of *AtCLCf* is provided in Supplementary Fig. 10. Recombinant protein expression and purification were carried out as described in detail in Invitrogen manual (https://assets.fishersci.com/TFS-Assets/LSG/manuals/easyselect_man.pdf?_ga=2.6716082.1874667851.1688092554-921422677.1688092554). In brief, the construct was introduced into SMD1168H *Pichia* yeast strain using electroporation method[65]. A colony with the highest protein expression level was selected for further analysis. The protein expression was induced in a 500 ml BMMY culture by adding 5 % methanol and growing the culture at 28 °C for 48 h. The cells were pelleted and resuspended in solubilization buffer (20 mM Tris-HCl, pH 8.0, 20 mM K₂HPO₄, 300 mM NaCl, 10% glycerol) and lysed with the use of glass beads. The unlysed cells were removed by centrifugation at 4 °C, 12,000 × *g* for 30 min and the supernatant was applied to a HisTrap HP column (GE healthcare), and the recombinant protein was eluted by adding 500 mM imidazole. Protein expression and purity were confirmed by SDS-PAGE and western-blot analyses using anti-His (Cat # SC−8036, 1:5000) and anti-Mouse (Cat # SC-2005, 1:30000) antibodies (Santacruz) (Supplementary Fig. 11).

Preparation of AtCLCf-liposomes and spectrofluorimetric analysis: Liposomes were prepared by the film rehydration method. 25 mg of soy lipid extract was dissolved in 2 ml of chloroform. The chloroform was evaporated using a rotary evaporator in a round-bottomed flask. The dried lipid film was rehydrated with 2 ml of 5 mM sodium phosphate buffer pH 7.5, plus chloride sensitive dye MQAE. Liposomes with uniform size were obtained by passing the dye incorporated liposomes through a hand-held, extruder with polycarbonate nucleopore filters (200 nm, Whatman). The entire 2 ml solution after the extrusion process was transferred to a dialysis cassette (Thermo Scientific) and dialyzed against 5 mM sodium phosphate buffer pH 7.5 for 2 days at 4 °C to remove the excess dye. The liposome suspension was removed from the dialysis cassette and used for AtCLCf incorporation. AtCLCf-liposomes were prepared by incubating the purified AtCLCf and dye incorporated liposomes (protein-to-lipid ratio of 1:100) for 15 min at room temperature. The spectrofluorimetric analysis for Cl⁻ transport was carried out as described for the PM vesicles above. For the HPTS assay, dried lipid was rehydrated with 2 ml of 10 mM HEPES buffer pH 7.0, 100 mM NaCl plus a pH-sensitive dye HPTS (8-hydroxy-1,3,6-pyrenetrisulfonate) The HPTS-containing liposome suspension was added to the buffer (10 mM HEPES, 100 mM NaX, pH 7.0, X = Cl⁻, F⁻, I⁻, NO₃⁻) for anion selectivity assays. A pH gradient of 7–8 was applied across the vesicles by addition of NaOH, and ion transport activities were monitored by increment of HPTS fluorescence (λex = 450 nm, λem = 510 nm) over 150 s. The HPTS assays were carried out from three such independent liposome preparations.

## Measurement of Cl⁻ content in the Arabidopsis protoplasts

Leaf mesophyll protoplasts were isolated from one-month-old WT, *atclcf*, *35S::AtCLCf;atclcf* and *35S::MD2AtCLCf;atclcf* (*35S::MD2*) seedlings. Protoplasts treated with 50 mM NaCl for 3 h were loaded with 100 μM MQAE for 30 min and visualized under confocal microscope (Fv3000). The percentage of protoplasts showing no quenching was calculated after deducting the number of protoplasts showing no fluorescence quenching by the total number of protoplasts in a microscopic field, 10 such microscopic fields were analyzed. In addition, the extent of fluorescence quenching (equivalent to the concentration of Cl⁻ retained inside the protoplasts) was measured from protoplasts treated with 50 mM NaCl for 3 h followed by 100 μM MQAE loading for 30 min. Assay was started with 2 ml of the suspension (500 μl of the protoplasts in 5 mM phosphate buffer). Triton X-100 (20 %) was added after 75 s and the quenching was recorded. The difference in the fluorescence value (final-initial value before adding Triton X-100) was plotted as quenched fluorescence.

## RNA isolation and quantitative real-time PCR (qRT-PCR) analysis

RNA was isolated from Arabidopsis using TRIzol reagent (Thermo Fisher) following the manufacturer's instructions[54]. From this, 1 μg of RNA was used and cDNA was synthesized using Maxima first strand cDNA synthesis kit for qRT-PCR (Thermo Fisher) following the manufacturer's instructions. The qRT-PCR analyses were performed using StepOne™ Real-Time PCR machine (Applied Biosystems, Foster City, CA, USA) with the following program: 20 s at 95 °C followed by 40 cycles of 03 s at 95 °C and 30 s at 60 °C using SYBR Fast ABI Prism PCR kit from KAPA (Biosystems, Wilmington, MA, USA). The qRT-PCR data were analyzed using the StepOne™ Software (v2.1, ABI). The primers used for the qRT-PCR analysis are listed in Supplementary Table S2. Gene expression levels from three biological replicates each with three technical replicates (total *n* = 9) were calculated based on ΔΔCT values and represented as relative expression levels (fold change) to constitutively expressed internal control, *AtUbiquitin 10*.

## Histochemical GUS staining

For histochemical study, transcriptional reporter line *pAtCLCf::GUS* was generated in WT background. To carry out GUS staining assays,

cold stratified seeds were sown on MS plates. One-week-old, untreated, and salt treated (100 mM NaCl for 6 h) seedlings as well as four-week-old seedling tissues were stained in the GUS staining solution [0.1 M sodium phosphate buffer pH 7.0, 10 mM EDTA, 0.1% Triton X-100, 2 mM 5-bromo-4-chloro-3-indolyl glucuronide (X-Gluc)] for 5 min followed by overnight incubation in the dark at 37 °C without shaking. After removing staining solution several washes with 50 % ethanol were performed to remove chlorophyll until the tissues were cleared[66]. The images of stained whole seedlings as well as various tissue parts were recorded using an Axiocam 506 color microscope (Zeiss). GUS expression in different parts of seedlings was quantified based on the relative intensities of blue coloration using ImageJ software (https://imagej.nih.gov/ij/download.html). Data presented are mean ± SE of three biological replicates, each biological replicate consisting of at least six plants. Statistical significance was determined by unpaired Student's *t* test (two-tailed).

### Estimation of total Cl⁻ ion concentration from plants

Control and salt-treated 4-week-old Arabidopsis plants were harvested and rinsed briefly with distilled water to remove surface contaminating Cl⁻. Pools of four plants were taken as one replicate, and three independent replicates were used to generate the mean values reported. Leaves and roots from plants were separated at collection and left to dry at 50 °C for 2 days. For determination of Cl⁻, the extracts were prepared by grinding 0.2 g of dry tissue with 10 ml of distilled water by incubating at 25 °C for 10 min. The homogenate was centrifuged at $3000 \times g$ for 15 min, and the supernatant filtered through Whatman qualitative filter paper (110 mm). An aliquot of filtrate was used for Cl⁻ estimation. The Cl⁻ concentration was measured by anion chromatography (820 IC Separation Centre; Metrohm Ion Chromatography System, Riverview, FL, USA) and presented as mg g⁻¹DW.

For Cl⁻ efflux analysis, 10-day-old seedlings of WT, *atclcf* and *35S::AtCLCf;atclcf* were treated with 100 mM NaCl in 5 ml of ½ MS liquid medium for 6 h (NaCl was not added for untreated seedlings), after which they were transferred to 5 ml of plain ½ MS liquid medium and allowed to stand for another 6 h. 2 ml of this liquid medium was taken and used to quantify the amount of Cl⁻ effluxed into the external medium by the Arabidopsis roots. One biological replicate in each genotype consisted of 15 seedlings and five such independent biological replicates were used for the analyses.

### Yeast strains and yeast complementation assay

*Saccharomyces cerevisiae* strain BY4741 (*MATa his3Δ1 leu2Δ met15Δ ura3Δ*; EUROSCARF) and its derivative: BY4741 (*MATa;ura3Δ0; leu2Δ0; his3Δ1; met15Δ0; YJRO40w::kanMX4*) were used to carry out the complementation experiments. Three selectivity filter domain binding sites were mutated by site-directed mutagenesis [(MD1(I204F, Q207Y), MD2(E247W) and MD3(L463W, P466F)] (Supplementary Fig. 12). The coding sequence of *AtCLCf* and mutated *AtCLCf* sequences were cloned downstream of the respective promoter into the yeast multicopy vector YEp352 and the primers used for amplification are listed in Supplementary Table 2. WT transformed with empty *YEp352*[67] plasmid was used as control. The yeast complementation assays with cells expressing *AtCLCf* were performed on semi-solid YNB medium containing appropriate supplements. Drop tests on YNB semi-solid medium with varying concentrations of NaCl (0.5, 1, and 1.25 M) were carried out as described earlier[53].

### Cell culture and transfection

Human Embryonic Kidney 293 (HEK293FT) cells were cultured in DMEM medium supplemented with 10 % fetal bovine serum and 1% penicillin and 1% streptomycin and maintained in 5% CO₂ incubator at 37 °C. For transfection, cells will be seeded on the petri dishes and grown overnight. Subsequently, 1.5 μg of pIRES2-EGFP or pIRES2-EGFP-HA-AtCLCf plasmids were transfected into the cells by lipofectamine

2000. The transfected cells were incubated for 24 h in a 5% CO₂ incubator at 37 °C. 24 h post transfection, the cells were split and seeded on coverslips pre-coated with poly-D lysine one day before recording.

### Whole-cell patch-clamp recordings and data analysis

The cells were recorded with the internal solution (pipette solution) containing (in mM) 140 CsCl, 5 EGTA, 10 HEPES, 1 MgCl₂, 0.5 Na₃GTP, 4 Mg-ATP, 10 Na-phoshocreatine pH 7.4 (adjusted with KOH) and external solution containing (in mM): 10 Glucose, 125 NaCl, 25 NaHCO₃, 1.25 NaH₂PO₄.2H₂O, 2.5 KCl, 1.8 CaCl₂, 1 MgCl₂, pH 7.4 (300–310 mOsm). To test the effect of internal and external pH on the chloride transport, in some experiments the internal or external pH was changed to pH5.5. To replace internal CsCl with CsF, CsBr, CsI, CsNO₃ or CsAcetate, the internal solution (pipette solution) contains (in mM) 130 CsF, CsBr, CsI, CsNO₃ or CsAcetate, 10 CsCl, 5 EGTA, 10 HEPES, 1 MgCl₂, 0.5 Na₃GTP, 4 Mg-ATP, 10 Na-phosphocreatine pH 7.4 (adjusted with CsOH). Whole cell recordings were performed with Multiclamp 700b amplifier (Molecular Device), low-pass filtered at 1 kHz and the series resistance was typically <10 MΩ after >50% compensation. Data was analyzed by Clampfit and Prism 10.

### Western blot analysis

To confirm the expression of HA-AtCLCf (Supplementary Fig. 13), transfected HEK293FT cells were lysed with RIPA lysis buffer at 4 °C. RIPA buffer consists of 150 mM sodium chloride, 0.1% Trion X-100, 0.5% sodium deoxycholate, 0.1% SDS and 50 mM Tris, pH 8.0 supplemented with mini, EDTA-free protease inhibitor cocktail (Roche). 30 μg of protein lysate was loaded for SDS-PAGE and transfer. Rabbit (rb) anti-HA antibody (Sigma Aldrich, Cat# H6908) was used in dilution of 1:1000 and β-actin was stained with anti-β-Actin (Sigma-Aldrich, A2228) antibody in the dilution of 1:5,000 as loading control.

### Statistical analysis

Data presented are the mean values ± SE or SD (as stated in the legends). Significance of difference among multiple samples was estimated by one-way ANOVA followed by Tukey's test. Means with different letters are significantly different, $P > 0.05$. Binary comparisons of data were statistically analyzed by unpaired, two-tailed Student's *t* test ($P < 0.05$ and $P < 0.01$).

### Reporting summary

Further information on research design is available in the Nature Portfolio Reporting Summary linked to this article.

## Data availability

All data are available in the main text or the supplementary materials. Source data are provided with this paper.

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

## Acknowledgements
The authors thank Drs. Akihiko Nakano and Tomohiro Uemura (RIKEN and Ochanomizu University, Japan) for providing plant material (seeds of GFP-RABA1b$^{Q72L}$ GFP-RABA1b$^{S27N}$), Dr. Prakash Arumugam (SIFBI, A*STAR, Singapore) for providing the yeast strains used in this study, and Dr. Jobichen Chacko for help with homology model building. We thank Prof. Elliot Meyerowitz (Caltech) and Dr. On Sun Lau (NUS) for critical reading of our manuscript. The National University of Singapore provided partial financial support as grant number A–8000149-03-00, and PhD research scholarship to S.R.

## Author contributions
P.K., S.R. and P.P.K. conceived the research plans; all authors helped design the experiments. J.F. designed the BEX5 experiments, J.X. designed the fine phenotyping experiments. H.H. and D.Y. carried out electrophysiological experiments. S.R. and P.K. carried out all other experiments and analyzed the data. P.K. and S.R. wrote the article with contributions from all authors.

## Competing interests
The authors declare no competing interests.
