## [Peer Review File · Nature Communications]

The translocation of a chloride channel from the Golgi to the plasma membrane helps plants adapt to salt stressReviewer #1 (Remarks to the Author):

Rajappa et al. reported that the chloride channel AtCLCf is translocated from the Golgi apparatus to the PM under salt stress to overcome Cl⁻ toxicity in plants. The authors also demonstrated that the WRKY transcription factor regulates AtCLCf expression under salt stress and that AtCLCf-deficient seedlings are more sensitive to salt. Since little is known about the molecular and cellular mechanisms of plant tolerance to Cl⁻ ion toxicity, this manuscript is quite interesting and includes several novel findings. However, I felt that the data presented here are insufficient to support the conclusions, and a data of higher quality is required for publication.

Major points

1. According to another study, AtCLCf is a TGN-localized antiporter (Scholl et al., 2021, JCS). The data and conclusion presented in this manuscript are quite different from those of Scholl's reports. I suggest that the authors thoroughly explain these disparities.

2. The quality of the images in Fig. 5 and Fig. 6 is quite low. These images do not adequately support the translocation of AtCLCf from the Golgi apparatus to the PM. It is unclear whether the destination of CLCf under salt stress conditions changes to the PM. Additionally, it appears that AtCLCf is localized on small dot structures, which are located in the peripheral regions of the PM. The primary location of AtCLCf in untreated cells does not appear to be the Golgi or TGN (Fig.6D). In addition, GFP/RFP-AtCLCf is expressed under the control of the 35S promoter, which induces overexpression. Therefore, to monitor subcellular localization in detail, the authors should express AtCLCf under the control of the native promoter.

3. To confirm the increasing Cl⁻ efflux, the experiments in Fig. 3A and B should be performed with and without NaCl.

4. The authors mentioned that the localization of AtCLCf shifts from the Golgi through TGN to the PM in the results section. However, in the discussion section, they claim that the localization of AtCLCf on the TGN/endosome is quite important. The distinction between Golgi and TGN seemed confusing to me as a reader.

5. Why is the Cl⁻ content of the treated root lower in the *atclcf* mutant and higher in the overexpression lines in Fig. 2M? This result appears contradictory considering the other findings of this work, which indicate that AtCLCf is increased in the root by salt treatment and that it appears to function as the effluxing carrier.

6. Quantification of AtCLCf colocalization with Golgi, TGN, or PM in Fig. 5 markers at 0, 4, and 6h NaCl treatment will validate the author's hypothesis.

7. The authors stated in the text that "root elongation was reduced in seedlings treated with 5 μ M BFA vs. controls not treated with BFA (Fig. 6C), mimicking the phenotype of *atclcf* roots"; however, Fig. 1 shows that *atclcf* roots are not shorter than WT without salt treatment. Therefore, if the authors intend to show the phenotype produced by impaired AtCLCf transport, I think salt treatment must be performed in combination with BFA.

Additionally, BFA induces other membrane traffic abnormalities, and is known to affect the auxin-dependent development of primary roots (Okumura et al., 2013, Plant Cell Physiol). The result as "shorter roots" does not necessarily reflect only the blocked translocation of AtCLCf.

8. It has been reported that RABA1 group members have redundant functions, and *raba1b/bex5* single mutants do not show any phenotype under normal growth conditions. However, as shown in Fig. 6C and E, the *bex5* roots appeared shorter without treatment. Do the authors have any explanation for this observation?

Minor Points

1. The last line: I suppose "Fig. 2o" is a typing error for Fig. 2M.

2. "...AtCLCf expression was significantly suppressed ~10-fold) in *atwrky9*, but not in *atwrky6*, *atwrky9*" The second "*atwrky9*" would be *atwrky33*(page 5).

3. Was AtWRKY9 expressed under pAtCLCf in Fig. 2o? It would be preferable to provide more

context for this analysis.

4. Since AtCLCf is a transmembrane protein, it would be misleading to draw the protein as if it is solubilized in the cytosol at the "Translocation" step in Fig.5E and 6G.

5. "AtCLCc and AtCLCg are... known to improve salt tolerance of plants." "in" is not necessary (page 9, line 4).

6. There are several extra periods following citation numbers (e.g., page 9, line 14).

7. The authors must demonstrate the extent of CLCf overexpression in 35S::CLCf transgenic lines compared to wild type plants.

8. How was the recovery experiment performed? Also, were the plants transferred to agar plates containing NaCl or submerged in a solution containing NaCl? M&M needs further explanation to ensure reproducibility of the experiments.

9. It has already been reported that expression of SN mutant of RABA1b affects the plant response to the salinity stress (Asaoka et al., Plant J, 2013). The novelty of the data shown in Figure 5E and 5F needs to be carefully stated.

10. The method for ChIP-qPCR should be mentioned in more detail in M&M section.

11. "AoWRKY" is typo on page 9.

Reviewer #2 (Remarks to the Author):

This is a very interesting paper on the role of CLCs in plant salinity tolerance, an area of research that is relatively underexposed. Rajappa et al convincingly show that CLCf plays a significant role in lowering the Cl load to the shoot and that this function depends on trafficking of CLCf from Golgi to plasma membrane.

I have a number of comments:

-If the authors want to support their claim this is an important 'novel' mechanism some better evidence of its significance would be called for. Six hours @ 50 NaCl is not very physiological. The one week treatment is more informative but why does it not include the shoot FW and whole seedling FW (or DW). Whole plant biomass in response to longer treatments, using multiple NaCl levels and mature plants would also be very useful.

-Fig 2 J-L: using 'shoot length' as a proxy for salt tolerance is not a good idea; there may very well be pleiotropic effects of CLCf on flowering which are independent of salinity. Much better to use FW or DW biomass during vegetative growth only.

-A major issue for functional characterisation of any transporter is its substrate selectivity. Fig 3 only looks at Cl; why not other anions, particularly nitrate? Using the vesicle quenching assays this could easily be done with a range of Na-salts to assess selectivity. This is especially important given the propensity of several CLCs to be nitrate selective.

-The quenching assay data do not say anything about 'directionality'. Firstly, in fig 3A the start of the trace is unclear (it should be as shown in 3B) and the KO line seems to show considerable quenching (~100 units) which (a) is much bigger than that in the iso vesicles (suggesting leak in the rso prep is much bigger) and (b) it is almost as much as shown for wt iso vesicles. In general, these assays cannot be compared quantitatively, especially since the fraction of rso versus iso is unknown, different membrane potentials can develop, background leak will vary, etc, etc. Furthermore, the authors state that 'There were no significant differences in the transport rates among RSO vesicles of different genotypes' but is this really the case? Fig 3A suggests the OX line quenching is ~130 units whereas that of the KO line is ~100. Potentially a significant difference. At any rate, data should be averaged across at least 3 separate vesicle preps. Thus, based on the fig 3 data I would not say there is good evidence that transport is 'directional'. Furthermore, 'directionality' would be hard to explain mechanistically if CLCf behaves like a channel. What is far more relevant is the flux in intact tissue and therefore the electrochemical gradient of Cl that pertains near the channel. That will decide the direction of any catalysed flux. The authors should provide estimates of some credible values for cytosolic and apoplasmic Cl levels and typical

membrane potentials and then calculate whether the gradient is inward or outward directed.

-The increase in CLCf expression (fig 2) appears to be transient and only lasts a few hours; please discuss the functional implications of this (fig 2)

-Were there any germination phenotypes in *clcf*, e.g. delayed germination? And if so, were seedling phenotypes corrected for this?

-Fig 4: again I have some trouble accepting the validity of these 'transport assays', especially the way they are 'quantified' (fraction of cells with fluorescence). What cut off was used for the fluorescence level? Are you sure the MQAE loading was the same in all preps? It would probably be much better to take a fixed volume of protoplast suspension and measure total fluorescence in a fluorimeter. This would allow a (calibrated) fluorescence signal per mg protein.

-'And ectopic expression of WRKY25, 33 improved salt tolerance' not clear what is meant here. Are Arabidopsis WRKYs referred to here? The refs relate to other species and isoforms (i.e. WRKY41 and WRKY5

-'Although salinity stress is one of the most extensively studied abiotic stresses in plants, the effects of counter-anions such as Cl⁻ are often neglected' sentence does not make sense; counter ions to what?

Reviewer #3 (Remarks to the Author):

In this work, Rajappa and co-authors investigate the role of CLC chloride channel in salinity tolerance in Arabidopsis. The authors report that AtCLCf expression is regulated by WRKY transcription factor and show that AtCLCf overexpression confers an enhanced resistance to salt stress. It is also shown that salt stress induced the translocation of GFP-AtCLCf fusion protein to PM where AtCLCf functions as a Cl⁻ efflux channel. Overall, it is concluded that the translocation of a chloride channel from the Golgi to the PM represents a novel mechanism of salt adaptation in plants.

Overall, I have found this work being rather competent and potentially interesting. However, from my personal view, the both significance and novelty of this work are a "borderline case". I also believe that this work lacks a few critical experiments that will strengthen the authors' arguments. My specific comments are as follow:

1. Both cytosolic chloride toxicity and essentiality of reducing cytosolic Cl content to confer normal plant operation are both common knowledge and hardly surprising. Similarly, CLC represent the major (and, arguably, only) pathway for Cl⁻ exclusion from the cell. Both these topics have been extensively covered in recent experimental papers and reviews (e.g., Geilfus 2018 PCP) so this work adds little to the current knowledge.

2. The authors advocate that endocytosis-mediated translocation of CLC from Golgi to PM is essential for salinity stress tolerance. Playing devil's advocate, I can argue that plants can employ a different and more effective strategy to achieve this goal, namely reducing amount of Cl⁻ taken by roots. Chloride uptake by roots is a thermodynamically ACTIVE process (see Bazihizina et al (2019) TIPS for details), even under conditions of soil salinity. So, why can't plants simply slow/shut down operation of H⁺/Cl⁻ symporters involved in this uptake? The possible answer could be that, in this case, plants may compromise uptake of other essential anions such as nitrate or phosphate. This calls for additional experiments involving silencing H/Cl symporter operation followed by phenotyping experiments and quantification of plant N and P content.

3. The conclusion that AtCLC functions as a directional Cl⁻ channel needs to be validated in electrophysiological experiments. This should also include the characterization of its selectivity. Right now, this part fails to convince me.

4. The difference in roots length can hardly be used as a proxy for salt tolerance. Pot experiments are required, following a proper biomass assessment.

5. The longest salinity treatment in this work was for 1 week. During this period, osmotic component of the salt stress dominates. Oddly, no isotonic osmotic controls were included in this work. This also calls for some additional experiments.

We thank all the reviewers for their constructive comments which helped us to improve the paper substantially. Below are the detailed responses to the concerns raised by them and the changes made in the present revision.

Response to Reviewer comments:

Reviewer #1 (Remarks to the Author):

Rajappa et al. reported that the chloride channel AtCLCf is translocated from the Golgi apparatus to the PM under salt stress to overcome Cl⁻ toxicity in plants. The authors also demonstrated that the WRKY transcription factor regulates AtCLCf expression under salt stress and that AtCLCf-deficient seedlings are more sensitive to salt. Since little is known about the molecular and cellular mechanisms of plant tolerance to Cl⁻ ion toxicity, this manuscript is quite interesting and includes several novel findings. However, I felt that the data presented here are insufficient to support the conclusions, and a data of higher quality is required for publication.

Major points

1. According to another study, AtCLCf is a TGN-localized antiporter (Scholl et al., 2021, JCS). The data and conclusion presented in this manuscript are quite different from those of Scholl's reports. I suggest that the authors thoroughly explain these disparities.

Ans: In the earlier report, Scholl et al., 2021 have shown that the AtCLCf and AtCLCd colocalize in the TGN/EE and function redundantly in male gametophyte development.

Based on the derived amino acid structure, the authors predicted that the CLCf might be a H⁺/Cl⁻ antiporter, although no experimental evidence for the same was provided.

However, we show that AtCLCf is translocated to PM upon NaCl treatment which was not tested in the earlier reports. Also, our liposome-based assays with a pH sensitive dye, HPTS confirm that AtCLCf is a Cl⁻ channel and not an antiporter. We have revised the discussion part to include this.

2. The quality of the images in Fig. 5 and Fig. 6 is quite low. These images do not adequately support the translocation of AtCLCf from the Golgi apparatus to the PM. It is unclear whether the destination of CLCf under salt stress conditions changes to the PM. Additionally, it appears that AtCLCf is localized on small dot structures, which are located in the peripheral regions of the PM. The primary location of AtCLCf in untreated cells does not appear to be the Golgi or TGN (Fig.6D).

Ans: We have provided better quality images for Fig. 5 and Fig. 6D which are now Fig. 6 and Fig. 7D in the revised version. We have also plotted the quantification data for AtCLCf localization at the Golgi, TGN and PM along with the markers, which clearly shows that

about 60% more AtCLCf is translocated to PM upon 6 h of salt treatment compared to 0 h values.

In addition, GFP/RFP-AtCLCf is expressed under the control of the 35S promoter, which induces overexpression. Therefore, to monitor subcellular localization in detail, the authors should express AtCLCf under the control of the native promoter.

Ans: We agree that to monitor subcellular localization, it is best to use the GFP-ATCLCf expressed under its native promoter. The protoplast localization images provided in Fig. 6E show the localization of AtCLCf under native promoter (*pAtCLCf::GFP-AtCLCf*), which is similar to the localization observed for the *35S::GFP-AtCLCf*. Therefore, we have used *35S::GFP-AtCLCf* for our subsequent experiments. This is now described in the revised manuscript.

3. To confirm the increasing Cl⁻ efflux, the experiments in Fig. 3A and B should be performed with and without NaCl.

Ans: Data with and without NaCl are provided in Fig 3A and B which are now moved to supplementary data (S5A and B). We have new liposome-based data in Fig 3 now.

4. The authors mentioned that the localization of AtCLCf shifts from the Golgi through TGN to the PM in the results section. However, in the discussion section, they claim that the localization of AtCLCf on the TGN/endosome is quite important. The distinction between Golgi and TGN seemed confusing to me as a reader.

Ans: The results and discussion sections have been revised now to make it clearer.

5. Why is the Cl⁻ content of the treated root lower in the *atclcf* mutant and higher in the overexpression lines in Fig. 2M? This result appears contradictory considering the other findings of this work, which indicate that AtCLCf is increased in the root by salt treatment and that it appears to function as the effluxing carrier.

Ans: The experiment was repeated, and fresh data is provided in the revised version.

6. Quantification of AtCLCf colocalization with Golgi, TGN, or PM in Fig. 5 markers at 0, 4, and 6h NaCl treatment will validate the author's hypothesis.

Ans: Quantification of AtCLCf colocalization with the markers has been included in the revised Figure 6.

7. The authors stated in the text that "root elongation was reduced in seedlings treated with 5 μ M BFA vs. controls not treated with BFA (Fig. 6C), mimicking the phenotype of *atclcf* roots"; however, Fig. 1 shows that *atclcf* roots are not shorter than WT without salt treatment. Therefore, if the

authors intend to show the phenotype produced by impaired AtCLCf transport, I think salt treatment must be performed in combination with BFA.

Ans: As suggested, we have performed salt treatment in combination with BFA and the new data are provided (old Fig. 6C) which is now Fig. 7C.

Additionally, BFA induces other membrane traffic abnormalities, and is known to affect the auxin-dependent development of primary roots (Okumura et al., 2013, Plant Cell Physiol). The result as “shorter roots” does not necessarily reflect only the blocked translocation of AtCLCf.

Ans: We agree that the BFA blocks whole protein secretion and trafficking machinery affecting various other pathways including AtCLCf translocation. This part is included in the discussion in revised version.

8. It has been reported that RABA1 group members have redundant functions, and *raba1b/bex5* single mutants do not show any phenotype under normal growth conditions. However, as shown in Fig. 6C and E, the *bex5* roots appeared shorter without treatment. Do the authors have any explanation for this observation?

Ans: We are aware that the AtRABA1 single mutants have redundant functions and did not show any significant phenotypes under normal conditions (Asaoka et al., 2013) – probably, minor differences were not focused on in the earlier study. However, we have consistently seen that the *bex5* roots appear shorter than the wild type even under untreated conditions and at this point, we do not have an explanation as to why *bex5* demonstrates such a phenotype, other than the fact that it consistently shows the phenotype we have described.

Minor Points

1. The last line: I suppose “Fig. 2o” is a typing error for Fig. 2M.

Ans: The correction has been made.

2. “...AtCLCf expression was significantly suppressed ~10-fold) in *atwrky9*, but not in *atwrky6*, *atwrky9*” The second “*atwrky9*” would be *atwrky33*(page 5).

Ans: Yes, it was a typo on our part. The correction has been made.

3. Was AtWRKY9 expressed under pAtCLCf in Fig. 2o? It would be preferable to provide more context for this analysis.

Ans: The AtWRKY9 was cloned under 35S promoter. Figure 2O is revised to avoid the confusion.

4. Since AtCLCf is a transmembrane protein, it would be misleading to draw the protein as if it is solubilized in the cytosol at the “Translocation” step in Fig.5E and 6G.

Ans: The current figure 7G has been revised as suggested.

5. "AtCLCc and AtCLCg are... known to improve salt tolerance of plants." "in" is not necessary (page 9, line 4).

Ans: The suggested change has been made

6. There are several extra periods following citation numbers (e.g., page 9, line 14).

Ans: Extra periods have been removed.

7. The authors must demonstrate the extent of CLCf overexpression in 35S::CLCf transgenic lines compared to wild type plants.

Ans: The qRT-PCR data showing expression of *AtCLCf* in 35S::*AtCLCf* transgenic lines is provided in the supplemental figure (Fig. S7D).

8. How was the recovery experiment performed? Also, were the plants transferred to agar plates containing NaCl or submerged in a solution containing NaCl? M&M needs further explanation to ensure reproducibility of the experiments.

Ans: The methods section has been revised to include the above info.

9. It has already been reported that expression of SN mutant of RABA1b affects the plant response to the salinity stress (Asaoka et al., Plant J, 2013). The novelty of the data shown in Figure 5E and 5F needs to be carefully stated.

Ans: We have now included this reference while stating the results in revised figure 6E and F. We are providing the underlying mechanism by which *AtCLCf* acts, which was not known previously.

10. The method for CHIP-qPCR should be mentioned in more detail in M&M section.

Ans: The relevant methods section has been revised to elaborate this.

11. "AoWRKY" is typo on page 9.

Ans: We thank the reviewer for pointing this error, we have rectified it in the revised manuscript.

Reviewer #2 (Remarks to the Author):

This is a very interesting paper on the role of CLCs in plant salinity tolerance, an area of research that is relatively underexposed. Rajappa et al convincingly show that CLCf plays a significant role in lowering the Cl load to the shoot and that this function depends on trafficking of CLCf from Golgi to plasma membrane.

I have a number of comments:

-If the authors want to support their claim this is an important 'novel' mechanism some better evidence of its significance would be called for. Six hours @ 50 NaCl is not very physiological. The one week treatment is more informative but why does it not include the shoot FW and whole seedling FW (or DW). Whole plant biomass in response to longer treatments, using multiple NaCl levels and mature plants would also be very useful.

We thank the reviewer for the highly encouraging comments along with the critical points for improvement.

-Fig 2 J-L: using 'shoot length' as a proxy for salt tolerance is not a good idea; there may very well be pleiotropic effects of CLCf on flowering which are independent of salinity. Much better to use FW or DW biomass during vegetative growth only.

Ans: We agree that measuring the shoot length may not be the ideal parameter. We have measured the FW and DW now and provided the FW/DW ratio in the revised version (Fig. 2L).

-A major issue for functional characterisation of any transporter is its substrate selectivity. Fig 3 only looks at Cl⁻; why not other anions, particularly nitrate? Using the vesicle quenching assays this could easily be done with a range of Na-salts to assess selectivity. This is especially important given the propensity of several CLCs to be nitrate selective.

Ans: There is a Ser residue in the AtCLCf which has been shown to confer a Cl⁻ channel function. In addition, we have now carried out transport rate analysis in liposome-based assays using MQAE and HPTS which further confirm that AtCLCf is a Cl⁻ channel. The data is provided in the revised version (Fig. 3).

-The quenching assay data do not say anything about 'directionality'. Firstly, in fig 3A the start of the trace is unclear (it should be as shown in 3B) and the KO line seems to show considerable quenching (~100 units) which (a) is much bigger than that in the iso vesicles (suggesting leak in the rso prep is much bigger) and (b) it is almost as much as shown for wt iso vesicles. In general, these assays cannot be compared quantitatively, especially since the fraction of rso versus iso is unknown, different membrane potentials can develop, background leak will vary, etc, etc. Furthermore, the authors state that 'There were no significant differences in the transport rates among RSO vesicles of different genotypes' but is this really the case? Fig 3A suggests the OX line quenching is ~130 units whereas that of the KO line is ~100. Potentially a significant difference. At any rate, data should be averaged across at least 3 separate vesicle preps. Thus, based on the fig 3 data I would not say there is good evidence that transport is 'directional'. Furthermore, 'directionality' would be hard to explain mechanistically if CLCf behaves like a channel. What is far more relevant is the flux in intact tissue and therefore the electrochemical gradient of Cl that pertains near the channel. That will decide the direction of any catalysed flux. The authors should

provide estimates of some credible values for cytosolic and apoplastic Cl levels and typical membrane potentials and then calculate whether the gradient is inward or outward directed.

Ans: In the assays with the vesicles, the start of the traces are shown for former Fig. 3A and B, which are now revised based on new measurements, and moved to Supplemental figures S. 5A and B. We have now plotted the data from 4 replicates.

We agree with the reviewer that it is hard to explain the directionality of the channel. However, our data suggest (see below) that AtCLCf may function with directionality. Besides the transport assays from the ISO and RSO PM vesicles, we have carried out (1) protoplast-based fluorescence analysis (confocal microscopy) and included the data in the revised version. (2) Additionally, in a spectrofluorimetric quantification experiment, protoplasts from *atclcf* and the *MD2::atclcf* clearly showed increased quenching of MQAE, because they retained more Cl⁻ within the protoplasts compared to the protoplasts stably expressing *35S::AtCLCf* (which should be the result of AtCLCf exporting Cl⁻ ions outward). Hence, we have claimed the directional transport.

The HPTS assay of liposomes with recombinant AtCLCf helped to show that it does not transport H⁺ and hence strengthening the channel nature rather than antiporter nature.

-The increase in CLCf expression (fig 2) appears to be transient and only lasts a few hours; please discuss the functional implications of this (fig 2)

Ans: Generally, the gene expression in response to salt stress occurs in the initial few hours. And once the protein is translated, it may function for longer than the duration of gene transcription. Therefore, we may see a transient increase in gene expression lasting for a few hours upon salt treatment, which is sufficient to activate the downstream processes.

-Were there any germination phenotypes in *clcf*, e.g. delayed germination? And if so, were seedling phenotypes corrected for this?

Ans: To overcome the concern that potential delays in seed germination may contribute to the observed root length phenotype, we have germinated the seeds on plain MS medium for the first 3 days and then transferred the seedlings (similar root length) to fresh MS medium with 50 mM NaCl (salt treatment). We recorded the root length measurements 5 days after the transfer. The data (MS plates below) exhibited the same trend as before. Therefore, we have retained the original pictures in the manuscript.

3-day-old Arabidopsis seedlings on MS agar medium

Arabidopsis seedlings 5-days after transfer to MS agar with NaCl

-Fig 4: again I have some trouble accepting the validity of these 'transport assays', especially the way they are 'quantified' (fraction of cells with fluorescence). What cut off was used for the fluorescence level? Are you sure the MQAE loading was the same in all preps? It would probably be much better to take a fixed volume of protoplast suspension and measure total fluorescence in a fluorimeter. This would allow a (calibrated) fluorescence signal per mg protein.

Ans: The protoplast-based assays have been repeated as suggested and the new (quantified) data are provided (Fig. 5F).

-'And ectopic expression of WRKY25, 33 improved salt tolerance' not clear what is meant here. Are Arabidopsis WRKYs referred to here? The refs relate to other species and isoforms (i.e. WRKY41 and

WRKY5

Ans: This portion has been revised to make it clearer in the revised version.

-‘Although salinity stress is one of the most extensively studied abiotic stresses in plants, the effects of counter-anions such as Cl⁻ are often neglected’ sentence does not make sense; counter ions to what?

Ans: This has been edited to: ‘Studies on the effect of salinity stress on plants have mostly focussed on the involvement of Na⁺, while the counter-anions such as Cl⁻ are often neglected’.

Reviewer #3 (Remarks to the Author):

In this work, Rajappa and co-authors investigate the role of CLC chloride channel in salinity tolerance in Arabidopsis. The authors report that AtCLCf expression is regulated by WRKY transcription factor and show that AtCLCf overexpression confers an enhanced resistance to salt stress. It is also shown that salt stress induced the translocation of GFP-AtCLCf fusion protein to PM where AtCLCf functions as a Cl⁻ efflux channel. Overall, it is concluded that the translocation of a chloride channel from the Golgi to the PM represents a novel mechanism of salt adaptation in plants.

Overall, I have found this work being rather competent and potentially interesting. However, from my personal view, the both significance and novelty of this work are a “borderline case”. I also believe that this work lacks a few critical experiments that will strengthen the authors’ arguments.

Again, we thank this reviewer for the highly encouraging comments along with the critical points for improvement.

My specific comments are as follow:

1. Both cytosolic chloride toxicity and essentiality of reducing cytosolic Cl content to confer normal plant operation are both common knowledge and hardly surprising. Similarly, CLC represent the major (and, arguably, only) pathway for Cl⁻ exclusion from the cell. Both these topics have been extensively covered in recent experimental papers and reviews (e.g., Geilfus 2018 PCP) so this work adds little to the current knowledge.

Ans: We agree that Cl⁻ uptake and its homeostasis is extensively covered in Geilfus 2018 PCP and some of the earlier papers have shown that CLCs are “involved in salt tolerance”. They have not provided the underlying mechanism of action. In our paper, we show a “novel mechanism” of salt-induced AtCLCf translocation from the Golgi to the PM mediated by AtRABA1b/BEX5, leading to increased salt tolerance in Arabidopsis.

In addition, we provide experimental evidence to demonstrate the importance of several critical residues present in the selectivity filter domains for proper functioning of ATCLCf.

By carrying out liposome-based assays with recombinant AtCLCf protein (new data) we show that AtCLCf is a Cl⁻ channel and not a Cl⁻/H⁺ antiporter (it was earlier speculated to be an antiporter based on amino acid sequence analysis – no experimental data were provided).

Further, studies on molecular regulation of CLCs are lacking. We show that the transcription factor, AtWRKY9 binds to the promoter and regulates *AtCLCf* under salt stress.

2. The authors advocate that endocytosis-mediated translocation of CLC from Golgi to PM is essential for salinity stress tolerance. Playing devil’s advocate, I can argue that plants can employ a different and more effective strategy to achieve this goal, namely reducing amount of Cl⁻ taken by roots. Chloride uptake by roots is a thermodynamically ACTIVE process (see Bazihizina et al (2019) TIPS for details), even under conditions of soil salinity. So, why can’t plants simply slow/shut down operation of H⁺/Cl⁻ symporters involved in this uptake? The possible answer could be that, in this case, plants may compromise uptake of other essential anions such as nitrate or phosphate. This calls for additional experiments involving silencing H/Cl symporter operation followed by phenotyping experiments and quantification of plant N and P content.

Ans: While we agree with the reviewer that additional experiments silencing the H⁺/Cl⁻ symporter studies might be interesting in future, they are beyond the scope of the present study. We feel that our claim has been substantiated with the insertion mutants as well as site-directed mutation of critical residues (some mutations resulting in abrogation of Cl⁻ transport).

3. The conclusion that AtCLC functions as a directional Cl⁻ channel needs to be validated in electrophysiological experiments. This should also include the characterization of its selectivity. Right now, this part fails to convince me.

Ans: We agree that electrophysiological measurements will add value to the conclusions. However, we do not have the facilities to carry out such work. Instead, we have now carried out protoplast-based assays (confocal microscopy and spectrofluorimetric quantification),

which was also suggested by Reviewer2. The newly provided quantitative data strongly suggests that AtCLCf functions as a Cl⁻ efflux channel. Hence, we hope that the reviewer will agree that the new data are an acceptable alternative to address the directionality of transport. Additionally, we have performed liposome-based functional assays using the recombinant AtCLCf protein and have included the data in Fig. 3.

4. The difference in roots length can hardly be used as a proxy for salt tolerance. Pot experiments are required, following a proper biomass assessment.

Ans: We appreciate the thoughtful suggestion from the Reviewer (also by the Reviewer1). Accordingly, we have conducted new experiments and measured the FW/DW values (included in the revised version Fig. 2L).

5. The longest salinity treatment in this work was for 1 week. During this period, osmotic component of the salt stress dominates. Oddly, no isotonic osmotic controls were included in this work. This also calls for some additional experiments.

Ans: We have done the root length assay with mannitol treatment (osmotic control) and presented the new data (Fig. S1). We do not see any significant changes contributed by the mannitol-induced osmotic effect.

Reviewer #1 (Remarks to the Author):

I carefully read revised the manuscript by Rajappa et al. I felt that the manuscript is improved compared to original version. However, a few points should be addressed sincerely.

1. Unfortunately I still think that the imaging data are still not enough to convince that the AtCLCf is translocated to the PM upon salt treatment. From my eyes, it seems to localize to the adjacent area of the PM in Fig6 or to small dot-structures in close vicinity to the PM in Fig7. In addition, the translocation of AtCLCf is not presented under the control of the native promoter. The rebuttal "The protoplast localization images provided in Fig. 6E show the localization of AtCLCf under native promoter (pAtCLCf::GFP-AtCLCf), which is similar to the localization observed for the 35S::GFP-AtCLCf. Therefore, we have used 35S::GFP-AtCLCf for our subsequent experiments" is not enough to convince me, because protoplast expression is quite specific condition to observe the translocation of AtCLCf. The translocation of AtCLCf to the PM is one of the crucial points for the novelty of this paper. I think it has to be shown more carefully in detail.

2. In my review comment number 3 (about the former Fig. 3A and B), what I meant by "with and without NaCl" was to perform the Cl⁻ transport experiment with the vesicles of the PM extracted from untreated plants or NaCl treated plants. Currently all the experiments were done with the vesicles prepared from NaCl treated plants. If the Cl⁻ transport is increased with the vesicles of the PM extracted from the NaCl treated plants compared to those of the PM from untreated plants, it would support the authors' conclusion that AtCLCf is translocated from intracellular membranes to the PM upon salt stress. Additional cycloheximide treatment might help to exclude the effect of salt-induced AtCLCf expression.

3. About the root length analysis with BFA, it would require careful quantification and interpretation of untreated, BFA or NaCl alone, and BFA+NaCl treated WT and mutant plants since each of BFA or NaCl treatment alone makes the roots shorter by many reasons. In addition, atclcf mutant complemented by pAtCLCf::GFP-AtCLCf should be included.

Reviewer #2 (Remarks to the Author):

The authors have improved their manuscript and added more experimental data. Nevertheless, some of the main issues are not addressed or only partially. The three main issues are:

-Transporter substrate selectivity

A major issue for functional characterisation of any transporter is its substrate selectivity. I suggested to look in particular to nitrate since many CLCs have a high nitrate selectivity. The authors state in their response that 'There is a Ser residue in the AtCLCf which has been shown to confer a Cl⁻ channel function. In addition, we have now carried out transport rate analysis in liposome-based assays using MQAE and HPTS which further confirm that AtCLCf is a Cl⁻ channel.' This does not make much sense: Firstly, the quenching assays do NOT show CLCf is 'a Cl channel' (see below) and they cannot inform regarding ion selectivity if you only test Cl. If CLCf behaves like a H⁺ antiport, selectivity could easily be established using acridine orange assays (or any other pH dye such as BCECF or HPTS) and addition of different anion salts (e.g. NaCl vs NaNO₃). If it functions as a channel, patch clamping would be the more appropriate technique, however I appreciate the authors may not have the facilities and expertise to carry this out. However, one could measure Cl using MQAE as a function of (potentially) competing anions (e.g. NaNO₃).

Since none of these experiments has been carried out, the authors should refrain from over interpreting their data, state that CLCf selectivity is unknown and discuss its implications.

-Transporter mechanism

As pointed out by previous reviews, there is some good evidence that CLCf functions as an antiport (e.g. Scholl et al 2020; Dukic et al 2022) rather than a channel. Since the transport mechanism can impact on transporter function (and physiological role) it would be useful to know how CLCf works. However, this is not simple and requires quite sophisticated el. physiology to do it properly.

Even then, it will not necessarily give conclusive outcomes about physiological function (for example, in animal cells the same CLC can function as H⁺/Cl⁻ exchanger in endomembranes but as Cl⁻ channels when trafficked to the plasmamembrane (e.g. Jentsch, 2015).

Vesicle/liposome assays are not very suitable to differentiate between transport modes. The presented liposome data of fig 3c do not make much sense; liposomes are suspended in phosphate buffer and there is no Cl in the assay. Since in antiport the fluxes are coupled there cannot be any H movement if there is no Cl to transport.

Furthermore, although the yeast complementation yields some useful data about crucial residues, the successful complementation suggests that (at least in yeast) CLCf functions as an exchanger rather than a channel since yeast Gef1 most likely functions as an antiporter (e.g. Elbaz-Alon et al 2014; Stockbridge et al 2012; Zifarelli 2013).

Again, the authors should refrain from over interpreting their data rather than stating (L195) 'Collectively, these findings show that AtCLCf functions as a Cl⁻ efflux channel'.

-CLC 'directionality'

Similarly, the conclusion that (L195) 'AtCLCf functions as a Cl⁻ efflux channel' is greatly exaggerated and there appears to be a fundamental misconception about how transporter (CLCf in this case) properties derived from artificial systems (vesicles, liposomes, etc) inform potential physiological function in planta. The results only show how CLCf functions in the given experimental context. Multiple el. phys. studies have shown that CLC channels can catalyse both inward and outward Cl flux (e.g. De Angeli 2006 Nature; Costa et al 2012, J Physiol; Ludewig et al 1997 J Gen Physiol). As explained before, the actual transport direction in planta depends on the Cl gradient in combination with the membrane potential. For example, if Cl-cyt is 10 mM and Cl-out 100 mM, the Cl flux is inward when the membrane potential is more positive than -60 mV. The Cl flux will be outward if the membrane potential is more negative than -60 mV (values of around -60 mV are regularly observed in saline conditions). During vesicle transport assays, the direction of the Cl flux depends on the Cl gradient of the assay buffers and the membrane potential. (To better interpret findings the latter is typically 'set' at a specific value by using a K gradient in combination with valinomycin, something that is missing from the vesicle/liposome assays in this study). Vesicle/liposome assays will not tell us anything about the direction of the Cl flux catalysed in vivo! The authors should provide credible estimates of values for cytosolic and apoplastic Cl levels and typical membrane potentials. This will allow them to calculate whether Cl efflux can be passive (via a Cl channel) or has to be directly energised (e.g. via Cl:H antiport). It would also be very useful to measure the actual Cl efflux (this can be easily done by 'loading plants' with NaCl and then measure Cl appearing in the external medium). A reduced Cl efflux in the mutant vs wt and non-treated vs NaCl treated plants would greatly support of the authors' claim.

Figure S5 is unintelligible; the legend says vesicles are derived from wt, mutant and 35s genotypes but only show control, wt and mutant. Furthermore, in contrast to what it says in the text, there is clear quenching in panel 'a' (rso vesicles) which is comparable in magnitude to that shown in panel 'b' (iso vesicles). What the authors probably mean to say is that the difference between mutant and wt signal very small. However, there may be many reasons (other than genotype difference) to explain this and furthermore, the results leave us with a quenching signal in rso vesicles that is more than twice as large as the 'real' (i.e. subtracted) signal shown for the iso vesicles. This obviously makes for a very precarious and non-robust assay; a slight difference in leakiness between wt and mutant vesicle preps would explain this behaviour. (The authors should also discuss where the large rso-vesicle signal comes from).

Other issues:

-I suggested the authors should use biomass (FW and/or DW) to further assess growth phenotypes. They now have included a figure showing 'FW/DW ratios'. This is a pretty useless parameter! Actual FWs and/or DWs should be given.

-There is a general lack of methodological detail: normalisation of fluorescence in protoplasts (e.g. what is a 'fluorescing protoplast' ? Normalisation of vesicle/liposome assays? Use of gramicidin. Technical or biological repeats?

-Fig 5 d shows clearly less fluorescence (i.e. more quenching) in the mutant than in wt!

-Another area of concern is reproducibility; judging by the (very) small error bars on the quenching assays I am pretty sure the repeats were technical repeats (but this is not explained) and carried out with the same vesicle/liposome preparation. Averaged data from at least 3 independent vesicle preps should be shown.

-It would also be very informative to see some additional quenching assays using vesicles derived from non-salt grown plants, especially since the CLCf mutant is not a proper null mutant (it shows only 10 times lower expression).

Reviewer #3 (Remarks to the Author):

The authors have done a good job addressing most issues raised by reviewers. Although I would still prefer to see a direct electrophysiological support for some claims, I accept the reply and a justification for an alternative approach. Thus, I have no more critical comments about this work.

We thank the reviewers again for their critical comments which helped us to improve the paper further. Below are the detailed responses to the concerns raised by them.

Reviewer #1 (Remarks to the Author):

I carefully read revised the manuscript by Rajappa et al. I felt that the manuscript is improved compared to original version. However, a few points should be addressed sincerely.

1. Unfortunately I still think that the imaging data are still not enough to convince that the AtCLCf is translocated to the PM upon salt treatment. From my eyes, it seems to localize to the adjacent area of the PM in Fig6 or to small dot-structures in close vicinity to the PM in Fig7. In addition, the translocation of AtCLCf is not presented under the control of the native promoter. The rebuttal “The protoplast localization images provided in Fig. 6E show the localization of AtCLCf under native promoter (*pAtCLCf::GFP-AtCLCf*), which is similar to the localization observed for the *35S::GFP-AtCLCf*. Therefore, we have used *35S::GFP-AtCLCf* for our subsequent experiments” is not enough to convince me, because protoplast expression is quite specific condition to observe the translocation of AtCLCf. The translocation of AtCLCf to the PM is one of the crucial points for the novelty of this paper. I think it has to be shown more carefully in detail.

Ans: We have provided new images of the *35S::AtCLCf;atclcf* showing localization of GFP-AtCLCf under 6h NaCl treated conditions in Figure 6 and hope these images are satisfactory. After from checking the AtCLCf localization in leaf protoplasts of *pAtCLCf::GFP-AtCLCf*, we have also checked the localization in the roots of *pAtCLCf::GFP-AtCLCf* which shows GFP-AtCLCf expression in the root cortical and epidermal cell membranes upon NaCl treatment.

2. In my review comment number 3 (about the former Fig. 3A and B), what I meant by "with and without NaCl" was to perform the Cl⁻ transport experiment with the vesicles of the PM extracted from untreated plants or NaCl treated plants. Currently all the experiments were done with the vesicles prepared from NaCl treated plants. If the Cl⁻ transport is increased with the vesicles of the PM extracted from the NaCl treated plants compared to those of the PM from untreated plants, it would support the authors' conclusion that AtCLCf is translocated from intracellular membranes to the PM upon salt stress. Additional cycloheximide treatment might help to exclude the effect of salt-induced AtCLCf expression.

Ans: We have now carried out electrophysiology (patch clamp) experiments by expressing AtCLCf in Human Embryonic Kidney 293 cells and provided the data. We have used the PM vesicle assays only as supporting information.

3. About the root length analysis with BFA, it would require careful quantification and interpretation of untreated, BFA or NaCl alone, and BFA+NaCl treated WT and mutant plants since each of BFA or NaCl treatment alone makes the roots shorter by many reasons. In addition, *atclcf* mutant complemented by *pAtCLCf::GFP-AtCLCf* should be included.

Ans: The following are the images showing similar effect of NaCl and BFA treatment in *pAtCLCf::GFP-AtCLCf* seedlings.

Arabidopsis seedlings 5-days after transfer to MS agar with NaCl+BFA

Reviewer #2 (Remarks to the Author):

The authors have improved their manuscript and added more experimental data. Nevertheless, some of the main issues are not addressed or only partially. The three main issues are:

-Transporter substrate selectivity

A major issue for functional characterisation of any transporter is its substrate selectivity. I suggested to look in particular to nitrate since many CLCs have a high nitrate selectivity. The authors state in their response that ‘There is a Ser residue in the *AtCLCf* which has been shown to confer a Cl⁻ channel function. In addition, we have now carried out transport rate analysis in liposome-based assays using MQAE and HPTS which further confirm that *AtCLCf* is a Cl⁻ channel.’ This does not make much sense: Firstly, the quenching assays do NOT show *CLCf* is ‘a Cl⁻ channel’ (see below) and they cannot inform regarding ion selectivity if you only test Cl⁻. If *CLCf* behaves like a H⁺ antiport, selectivity could easily be established using acridine orange assays (or any other pH dye such as BCECF or HPTS) and addition of different anion salts (e.g. NaCl vs NaNO₃). If it functions as a channel, patch clamping would be the more appropriate technique, however I appreciate the authors may not

have the facilities and expertise to carry this out. However, one could measure Cl using MQAE as a function of (potentially) competing anions (e.g. NaNO₃).

Ans: We have now addressed this concern by carrying out two experiments: 1) Patch clamp recording by expressing AtCLCf in Human Embryonic Kidney 293 (HEK293FT) cells and HPTS assays in liposomes and presented the data in Figure 3 and S6. Our new data shows that AtCLCf functions as a Cl⁻/H⁺ antiporter and has high selectivity for Cl⁻ compared to F⁻, I⁻, Br⁻, Acetate and does not show any transport with NO₃⁻.

Since none of these experiments has been carried out, the authors should refrain from over interpreting their data, state that CLCf selectivity is unknown and discuss its implications.

-Transporter mechanism

As pointed out by previous reviews, there is some good evidence that CLCf functions as an antiport (e.g. Scholl et al 2020; Dukic et al 2022) rather than a channel. Since the transport mechanism can impact on transporter function (and physiological role) it would be useful to know how CLCf works. However, this is not simple and requires quite sophisticated electrophysiology to do it properly. Even then, it will not necessarily give conclusive outcomes about physiological function (for example, in animal cells the same ClC can function as H⁺/Cl⁻ exchanger in endomembranes but as Cl⁻ channels when trafficked to the plasmamembrane (e.g. Jentsch, 2015).

Vesicle/liposome assays are not very suitable to differentiate between transport modes. The presented liposome data of fig 3c do not make much sense; liposomes are suspended in phosphate buffer and there is no Cl in the assay. Since in antiport the fluxes are coupled there cannot be any H movement if there is no Cl to transport.

Ans: We thank the reviewer for this suggestion. We have now carried out HPTS assays in liposomes by including 100 mM NaCl inside the liposome and the results show that AtCLCf functions as a Cl⁻/H⁺ antiporter. As mentioned above, we have additionally carried out electrophysiological measurements in HEK293FT cells and presented the data.

Furthermore, although the yeast complementation yields some useful data about crucial residues, the successful complementation suggests that (at least in yeast) CLCf functions as an exchanger rather than a channel since yeast Gef1 most likely functions as an antiporter (e.g. Elbaz-Alon et al 2014; Stockbridge et al 2012; Zifarelli 2013).

Ans: We agree with the reviewer that yeast complementation data also suggests that AtCLCf might function as an antiporter and our new liposome-based and electrophysiology data with support this.

Again, the authors should refrain from over interpreting their data rather than stating (L195) 'Collectively, these findings show that AtCLCf functions as a Cl⁻ efflux channel'.

Ans: This section has been revised now. Also, new electrophysiology data which shows that AtCLCf functions in Cl⁻ efflux is included (Fig. 3).

-CLC 'directionality'

Similarly, the conclusion that (L195) 'AtCLCf functions as a Cl⁻ efflux channel' is greatly exaggerated and there appears to be a fundamental misconception about how transporter (CLCf in this case) properties derived from artificial systems (vesicles, liposomes, etc) inform potential physiological function in planta. The results only show how CLCf functions in the given experimental context. Multiple el. phys. studies have shown that CLC channels can catalyse both inward and outward Cl flux (e.g. De Angeli 2006 Nature; Costa et al 2012, J Physiol; Ludewig et al 1997 J Gen Physiol). As explained before, the actual transport direction in planta depends on the Cl gradient in combination with the membrane potential. For example, if Cl-cyt is 10 mM and Cl-out 100 mM, the Cl flux is inward when the membrane potential is more positive than -60 mV. The Cl flux will be outward if the membrane potential is more negative than -60 mV (values of around -60 mV are regularly observed in saline conditions). During vesicle transport assays, the direction of the Cl flux depends on the Cl gradient of the assay buffers and the membrane potential. (To better interpret findings the latter is typically 'set' at a specific value by using a K gradient in combination with valinomycin, something that is missing from the vesicle/liposome assays in this study). Vesicle/liposome assays will not tell us anything about the direction of the Cl flux catalysed in vivo! The authors should provide credible estimates of values for cytosolic and apoplastic Cl levels and typical membrane potentials. This will allow them to calculate whether Cl efflux can be passive (via a Cl channel) or has to be directly energised (e.g. via Cl:H antiport). It would also be very useful to measure the actual Cl efflux (this can be easily done by 'loading plants' with NaCl and then measure Cl appearing in the external medium). A reduced Cl efflux in the mutant vs wt and non-treated vs NaCl treated plants would greatly support of the authors' claim.

Ans: To address the above concerns, we have now carried out detailed electrophysiology experiments as mentioned above and provided the data in Figure 3 and S6.

In addition, we have carried our Cl⁻ efflux assay as suggested by the reviewer and provided the data in Figure 5 and we do observe a significant differences in the Cl⁻ efflux between WT, *atclcf* and *35S::AtCLCf*.

Figure S5 is unintelligible; the legend says vesicles are derived from wt, mutant and 35s genotypes but only show control, wt and mutant. Furthermore, in contrast to what it says in the text, there is clear quenching in panel 'a' (rso vesicles) which is comparable in magnitude to that shown in panel 'b' (iso vesicles). What the authors probably mean to say is that the difference between mutant and wt signal very small. However, there may be many reasons (other than genotype difference) to explain this and furthermore, the results leave us with a quenching signal in rso vesicles that is more than twice as large as the 'real' (i.e. subtracted) signal shown for the iso vesicles. This obviously makes for a very precarious and non-robust assay; a slight difference in leakiness between wt and mutant vesicle preps would explain this behaviour. (The authors should also discuss where the large rso-vesicle signal comes from).

Ans: Legend to Figure S5 is now revised for more clarity. We hope that the new electrophysiology measurements will address these concerns, and the PM vesicle data is only used as a supplemental data for our claims.

Other issues:

-I suggested the authors should use biomass (FW and/or DW) to further assess growth phenotypes. They now have included a figure showing 'FW/DW ratios'. This is a pretty useless parameter! Actual FWs and/or DWs should be given.

Ans: As suggested by the reviewer, we have provided the FW values of untreated and treated plants in the supplementary table now.

-There is a general lack of methodological detail: normalisation of fluorescence in protoplasts (e.g. what is a 'fluorescing protoplast'? Normalisation of vesicle/liposome assays? Use of gramicidin. Technical or biological repeats?

Ans: The methods sections are revised now, and details are included.

-Fig 5 d shows clearly less fluorescence (i.e. more quenching) in the mutant than in wt!

Ans: This section has been revised for clarity.

-Another area of concern is reproducibility; judging by the (very) small error bars on the quenching assays I am pretty sure the repeats were technical repeats (but this is not explained) and carried out with the same vesicle/liposome preparation. Averaged data from at least 3 independent vesicle preps should be shown.

Ans: All the fluorescence quenching data provided now are from at least 3 biological replicates of vesicles and liposomes.

-It would also be very informative to see some additional quenching assays using vesicles derived from non-salt grown plants, especially since the CLCf mutant is not a proper null mutant (it shows only 10 times lower expression).

Ans: Considering the reservations expressed by the reviewers, we decide not to repeat the suggested vesicle assays, and instead moved the vesicle data to the Suppl file now. The new data from additional liposome assays and electrophysiological measurements take the main data space, which we hope will be satisfactory.

Reviewer #3 (Remarks to the Author):

The authors have done a good job addressing most issues raised by reviewers. Although I would still prefer to see a direct electrophysiological support for some claims, I accept the reply and a justification for an alternative approach. Thus, I have no more critical comments about this work.

Ans: We appreciate the supportive comments. We have also carried out electrophysiological measurements by expressing AtCLCf in Human Embryonic Kidney 293 (HEK293FT) cells and included the data in revised version (Fig 3 and S6).

Reviewer #1 (Remarks to the Author):

I have read the revised manuscript and responses carefully, and found that the authors addressed raised questions sincerely.

Reviewer #2 (Remarks to the Author):

The authors have improved their manuscript and added more experimental data. All main issues have been addressed satisfactorily (see below).

-Transporter substrate selectivity

-this issue has now been resolved; new (el. Phys) data show Cl-H antiport activity

-Transporter mechanism

-resolved; see above

-CLC 'directionality' and yeast complementation

-resolved: see above

-Figure S5

Legend has been improved and is now intelligible

Other issues:

-use biomass (FW and/or DW) to further assess growth phenotypes.

-this has now been included

-There is a general lack of methodological detail

-improved description of methodology, including details about reproducibility is now presented

-Additional quenching assays using vesicles derived from non-salt grown plants

Authors' answer: Considering the reservations expressed by the reviewers, we decide not to repeat the suggested vesicle assays, and instead moved the vesicle data to the Suppl file now. The new data from additional liposome assays and electrophysiological measurements take the main data space, which we hope will be satisfactory.

-resolved